# Generalizable Lightweight Proxy for Robust NAS against Diverse Perturbations

**Hyeonjeong Ha**[1*], **Minseon Kim**[1*], **Sung Ju Hwang**[1,2]
[1]Korea Advanced Institute of Science and Technology (KAIST), [2]DeepAuto.ai
{hyeonjeongha, minseonkim, sjhwang82}@kaist.ac.kr

## Abstract

Recent neural architecture search (NAS) frameworks have been successful in finding optimal architectures for given conditions (e.g., performance or latency). However, they search for optimal architectures in terms of their performance on clean images only, while robustness against various types of perturbations or corruptions is crucial in practice. Although there exist several robust NAS frameworks that tackle this issue by integrating adversarial training into one-shot NAS, however, they are limited in that they only consider robustness against adversarial attacks and require significant computational resources to discover optimal architectures for a single task, which makes them impractical in real-world scenarios. To address these challenges, we propose a novel lightweight robust zero-cost proxy that considers the consistency across features, parameters, and gradients of both clean and perturbed images at the initialization state. Our approach facilitates an efficient and rapid search for neural architectures capable of learning generalizable features that exhibit robustness across diverse perturbations. The experimental results demonstrate that our proxy can rapidly and efficiently search for neural architectures that are consistently robust against various perturbations on multiple benchmark datasets and diverse search spaces, largely outperforming existing clean zero-shot NAS and robust NAS with reduced search cost. Code is available at https://github.com/HyeonjeongHa/CRoZe.

## 1 Introduction

Neural architecture search (NAS) techniques have achieved remarkable success in optimizing neural networks for given tasks and constraints, yielding networks that outperform handcrafted neural architectures [2, 24, 28, 33, 42]. However, previous NAS approaches have primarily aimed to search for architectures with optimal performance and efficiency on clean inputs, while paying less attention to robustness against adversarial perturbations [16, 29] or common types of corruptions [20]. This can result in finding unsafe and vulnerable architectures with erroneous and high-confidence predictions on input examples even with small perturbations [31, 22], limiting the practical deployment of NAS in real-world safety-critical applications.

To address the gap between robustness and NAS, previous robust NAS works [31, 18] have proposed to search for adversarially robust architectures by integrating adversarial training into NAS. Yet, they are computationally inefficient as they utilize costly adversarial training on top of the one-shot NAS methods [26, 4], requiring up to $33\times$ larger computational cost than clean one-shot NAS [42]. Especially, Guo et al. [18] takes almost 4 GPU days on NVIDIA 3090 RTX GPU to train the supernet, as it requires performing adversarial training on subnets with perturbed examples (Figure 2, RobNet). Furthermore, they only target a single type of perturbation, i.e., adversarial perturbation [16, 29], thus, failing to generalize to diverse perturbations. In order to deploy NAS to real-world applications

---

[*]Equal contribution. Author ordering is determined by coin flip.

37th Conference on Neural Information Processing Systems (NeurIPS 2023).

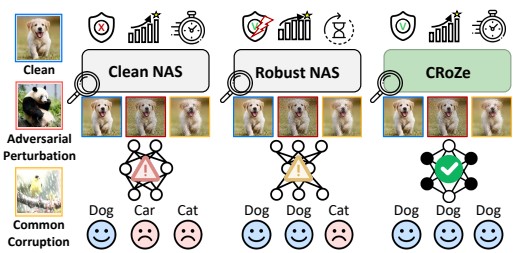

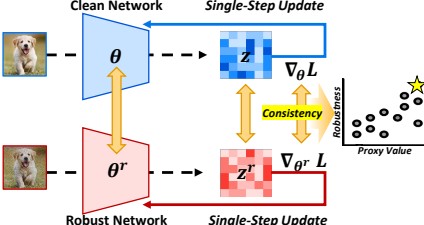

(a) Clean NAS and Robust NAS vs CRoZe.

(b) Consistency-based zero-cost proxy (CRoZe).

Figure 1: **Generalizable lightweight proxy for robust NAS against diverse perturbations.** While previous NAS methods search for neural architectures primarily on clean samples (Clean NAS) or adversarial perturbations (Robust NAS) with excessive search costs and fail to generalize across diverse perturbations, our proposed proxy, namely CRoZe, can rapidly search for high-performing neural architectures against diverse perturbations. Specifically, CRoZe evaluates the network's robustness in a single step based on the consistency across the features ($z$ and $z^r$), parameters ($\theta$ and $\theta^r$), and gradients ($\nabla_\theta \mathcal{L}$ and $\nabla_{\theta^r} \mathcal{L}$) between clean and robust network against clean and perturbed inputs, respectively.

that require handling diverse types of tasks and perturbations, we need a lightweight NAS method that can yield robust architectures without going over such costly processes.

To tackle this challenge, we propose a novel and lightweight **C**onsistency-based **Ro**bust **Ze**ro-cost proxy (**CRoZe**) that can rapidly evaluate the robustness of the neural architectures against *diverse semantic-preserving* perturbations without requiring any iterative training. While prior clean zero-shot NAS methods [1, 30] introduced proxies that score the networks with randomly initialized parameters [23, 39, 27, 37] without any training, they only consider which parameters are highly sensi-

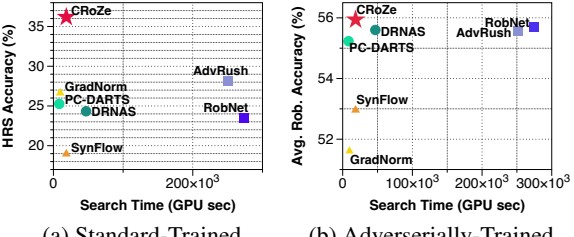

(a) Standard-Trained    (b) Adversarially-Trained

Figure 2: Final performance of the searched network in DARTS search space on CIFAR-10 through clean one-shot NAS, robust NAS, clean zero-shot NAS and our CRoZe.

tive to clean inputs for a given task, as determined by measuring the scale of the gradients based on the objectives and thus yield networks that are vulnerable against perturbed inputs (Figure 1a).

Specifically, our proxy captures the consistency across the features, parameters, and gradients of a randomly initialized model for both clean and perturbed inputs, which is updated with a single gradient step (Figure 1b). This metric design measures the model's robustness in multiple aspects, which is indicative of its generalized robustness to diverse types of perturbations. This prevents the metric from being biased toward a specific type of perturbation and ensures its robustness across diverse semantic-preserving perturbations. Empirically, we find that a neural architecture with the highest performance for a single type of perturbation tends to exhibit larger feature variance for other types of perturbations (Figure 4), while our proxy that considers the robustness in multiple aspects obtains features with smaller variance even on diverse types of perturbations. This suggests that our proxy is able to effectively discover architectures with enhanced generalized robustness.

We validate our approach through extensive experiments on diverse search spaces (NAS-Bench 201, DARTS) and multiple datasets (CIFAR-10, CIFAR-100, ImageNet16-120), with not only the adversarial perturbations but also with various types of common corruptions [20], against both clean zero-shot NAS [1, 30] and robust NAS baselines [31, 18]. The experimental results clearly demonstrate our method's effectiveness in finding generalizable robust neural architectures, as well as its computational efficiency. Our contributions can be summarized as follows:

- We propose a simple yet effective consistency-based zero-cost proxy for robust NAS against diverse perturbations via measuring the consistency of features, parameters, and gradients between perturbed and clean samples.

- Our approach can rapidly search for generalizable neural architectures that do not perform well only on clean samples but also are highly robust against diverse types of perturbations on various datasets.

- Our proxy obtains superior Spearman's $\rho$ across benchmarks compared to existing clean zero-shot NAS methods and identifies robust architectures that exceed robust NAS frameworks by **5.57%** with **14.7 times less search cost** within the DARTS search space on CIFAR-10.

## 2   Related Work

**Robustness of DNNs against Perturbations.**   Despite the recent advances, deep neural networks (DNNs) are still vulnerable to small perturbations on the input, e.g., common corruptions [20], random noises [10], and adversarial perturbations [3, 36], which can result in incorrect predictions with high confidence. To overcome such vulnerability against diverse perturbations, many approaches have been proposed to train the neural network to be robust against each type of perturbation individually. To learn a rich visual representation from limited crawled data, previous works [21, 8] utilized a combination of strong data augmentation functions to improve robustness to common corruptions and random Gaussian noises. Furthermore, to overcome adversarial vulnerability, widely used defense mechanisms [16, 29, 32] generate adversarially perturbed images by taking multiple gradient steps to maximize the training loss and use them in training to improve the model's robustness.

**Neural Architecture Search.**   Neural architecture search (NAS) leverages reinforcement learning [45, 2, 44] or evolutionary algorithms [34, 25, 13, 35] to automate the design of optimal architectures for specific tasks or devices. However, those are computationally intensive, making them impractical to be applied in real-world applications. To address this, zero-shot NAS methods [1, 30] have emerged that significantly reduce search costs by predicting the performance of architecture at the initialization state only with a single batch of a given dataset. Despite the improvement in NAS, previous zero-shot NAS methods, and conventional NAS methods aim only to find architectures with high accuracy on clean examples, without considering their robustness against various perturbations. In particular, SynFlow [1] lacks data incorporation in its scoring mechanism, potentially failing to find the network that can handle diverse perturbations. NASWOT [30] search models with low activation correlation across two different inputs, emphasizing their distinguishability. However, the criterion contradicts the requirements of robustness, which necessitates finding a network that maintains similar activations for both clean and perturbed inputs. As a result, models found with previous NAS methods often lead to incorrect predictions with high confidence [31, 22] even with small imperceptible perturbations applied to the inputs. A new class of NAS methods [18, 31] that considers robustness against adversarial perturbations has emerged. Yet, they require adversarial training of the supernet, which demands more computational resources than conventional NAS [34, 13] due to repeated adversarial gradient steps. Furthermore, adversarial robust NAS often overfit to a single type of perturbation due to only considering the adversarial robustness. Thus, there is a need for a lightweight NAS approach that can achieve generalized robustness for safe real-world applications.

## 3   Methods

Our ultimate goal is to efficiently search for robust architectures that have high performance on various tasks, regardless of the type of perturbations applied to the input samples. To achieve this goal, we propose a **C**onsistency-based **Ro**bust **Ze**ro-cost proxy (**CRoZe**) that considers the consistency of the features, parameters, and gradients between a single batch of clean and perturbed inputs obtained by taking a single gradient step. The proposed proxy enables the rapid evaluation of the robustness of the neural architectures in the random states, without any adversarial training (Figure 1).

### 3.1   Robust Architectures

Formally, our goal is to accurately estimate the final robustness of a given neural architecture $\mathcal{A}$ and a single batch of inputs $B = \{(x, y)\}$, without training. Here, $x \in X$ is the input sample, and $y \in Y$ is its corresponding label for the given dataset $\mathcal{D}$. In the following section, the network $f_\theta(\cdot)$ consists of an encoder $e_\psi(\cdot)$ and a linear layer $h_\pi(\cdot)$, which is $\mathcal{A}$ that is parameterized with $\psi$ and $\pi$, respectively. The straightforward approach to evaluating the robustness of the network is measuring the accuracy against the perturbed input $x'$ with unseen semantic-preserving perturbations, as follows:

$$\text{Accuracy} = \frac{1}{N} \sum_{n=1}^{N} \zeta(\arg\max_{c \in Y} \mathbb{P}(h_\pi \circ e_\psi(x') = c) = y), \tag{1}$$

where $\zeta$ is the Kronecker delta function, which returns 1 if the predicted class $c$ is the same as $y$, and 0 otherwise, $x'$ is a perturbed input, such as one with random Gaussian noise, common types of corruptions [21], or adversarial perturbations [29, 16] applied to it. Specifically, to have a correct prediction on the unseen perturbed input $x'$, the model needs to extract similar features between $x'$ and $x$, assuming that the model can correctly predict the label for input $x$ as follows:

$$\| e_\psi(x) - e_\psi(x') \| \le \epsilon, \tag{2}$$

where $\epsilon$ is sufficiently small bound. Thus, a robust model is one that can extract consistent features across a wide range of perturbations. However, precisely assessing the accuracy of the model against perturbed inputs requires training from scratch with a full dataset, which incurs a linear increase in the computational cost with respect to the number of neural architectures to be evaluated.

## 3.2 Estimating Robust Network through Perturbation

In this section, we explain details on preliminary protocols before computing our proxy. Due to the impractical computation cost to obtain a combinatorial number of fully-trained models in a given neural architecture search space (i.e., $10^{19}$ for DARTS), we propose to utilize two surrogate networks which together can estimate the robustness of fully-trained networks within a single gradient step. The two surrogate networks are a clean network $f_\theta$ with the randomly initialized parameter $\theta$ and a robust network $f_{\theta^r}$ with the robust parameter $\theta^r$, which is determined with a parameter perturbations from $f_\theta$. Then, the obtained $\theta^r$ is used to generate the single batch of perturbed inputs for our proxy.

**Robust Parameter Update via Layer-wise Parameter Perturbation.** We employ a surrogate robust network to estimate the output of fully-trained networks against perturbed inputs. To make the perturbation stronger, we use a double-perturbation scheme that combines layer-wise parameter perturbations [41] and input perturbations, both of which maximize the training objectives $\mathcal{L}$. This layer-wise perturbation allows us to learn smoother updated parameters by min-max optimization, through which we can obtain the model with the maximal possible generalization capability [41, 14] within a single step. Specifically, given a network $f$ is composed of $M$ layers, $f_\theta = f_{\theta_M} \circ \cdots \circ f_{\theta_1}$, with parameters $\theta = \{\theta_1, \ldots, \theta_M\}$, the $m^{th}$ layer-wise parameter perturbation is done as follows:

$$\theta_m^r \leftarrow \theta_m + \beta * \frac{\nabla_{\theta_m} \mathcal{L}\big(f_\theta(x), y\big)}{\|\nabla_{\theta_m} \mathcal{L}\big(f_\theta(x), y\big)\|} * \|\theta_m\|, \tag{3}$$

where $\beta$ is the step size for parameter perturbations, $\| \cdot \|$ is the norm, and $\mathcal{L}$ is the cross-entropy objective. This bounds the size of the perturbation by the norm of the original parameter $\|\theta_m\|$.

**Perturbed Input.** On top of the perturbed parameters (Eq. 3), we generate perturbed input images by employing fast gradient sign method (FGSM) [16], which is the worst case adversarial perturbation to the input $x$ as follows:

$$\delta = \epsilon \, \mathtt{sign}\Big(\nabla_x \mathcal{L}\big(f_{\theta^r}(x), y\big)\Big), \tag{4}$$

where $\delta$ is a generated adversarial perturbation that maximizes the cross-entropy objective $\mathcal{L}$ of given input $x$ and given label $y$. Then, we utilize the perturbed inputs ($x' = x + \delta$) to estimate the robustness of the fully-trained model. Although CRoZe is an input perturbation-agnostic proxy (Supplementary B.2), we employ adversarially perturbed inputs for all the following sections.

## 3.3 Consistency-based Proxy for Robust NAS

We now elaborate the details on our proxy that evaluate the robustness of the architecture with the two surrogate networks: the clean network that is randomly initialized and uses clean images $x$ as inputs, and the robust network parameterized with $\theta^r$ which uses perturbed images $x + \delta$ as inputs.

**Features, Parameters, and Gradients.** As we described in Section 3.1, we first evaluate the representational consistency between clean input ($x$) and perturbed input ($x'$) by forwarding them through the encoder of clean surrogate network $f_\theta(\cdot)$ and robust surrogate network $f_{\theta^r}(\cdot)$, respectively, as follows:

$$\mathcal{Z}_m(f_\theta(x), f_{\theta^r}(x')) = 1 + \frac{z_m \cdot z_m^r}{\|z_m\|\|z_m^r\|}, \tag{5}$$

where $z_m$ and $z_m^r$ are output feature of each network $f_\theta(\cdot)$ and $f_{\theta^r}(\cdot)$, respectively, from each $m^{th}$ layer. Especially, we measure layer-wise consistency with cosine similarity function between clean and robust features. The higher feature consistency infers the higher robustness of the network.

However, the proxy solely considering the feature consistency within a single batch can be heavily reliant on the selection of the batch. Therefore, to complement the feature consistency, we propose incorporating the consistency of updated parameters and gradient conflicts from each surrogate network as additional measures to evaluate the robustness of the network. To introduce these concepts, let us first denote the gradient and updated parameter of each surrogate network. The gradient $g$ from the clean surrogate network $f_\theta$ and robust surrogate network $f_{\theta^r}$ against clean images $x$ and perturbed images $x'$, respectively, are obtained as follows:

$$g = \nabla_\theta \mathcal{L}\big(f_\theta(x), y\big), \quad g^r = \nabla_{\theta^r} \mathcal{L}\big(f_{\theta^r}(x'), y\big), \tag{6}$$

where $g$ and $g^r$ are the gradients with respect to cross-entropy objectives $\mathcal{L}$ for clean images $x$ and perturbed images $x'$, respectively. Then, we can acquire single-step updated clean parameters $\theta$ and robust parameters $\theta^r$ calculated with gradients $g$ and $g^r$ and learning rate $\gamma$, respectively as follows:

$$\theta_1 \leftarrow \theta - \gamma g, \quad \theta_1^r \leftarrow \theta^r - \gamma g^r. \tag{7}$$

Since each surrogate network represents the model for each task, i.e., clean classification and perturbed classification, the parameters and gradients of each surrogate network correspond to the updated weights and convergence directions for each task. Thus, the network that has high robustness will exhibit identical or similar parameter spaces for both classification tasks. However, as acquiring parameters of a fully-trained network is impractical, we estimate the converged parameters with the single-step updated parameters $\theta_1$ and $\theta_1^r$. Accordingly, since the higher similarity of single-step updated parameters may promote the model to converge to an identical or similar parameter space for both tasks, we evaluate the parameter similarity as one of our proxy terms as follows:

$$\mathcal{P}_m(\theta_1, \theta_1^r) = 1 + \frac{\theta_{1,m} \cdot \theta_{1,m}^r}{\|\theta_{1,m}\|\|\theta_{1,m}^r\|}. \tag{8}$$

Moreover, each gradient of the surrogate networks represents the converged direction of given objectives for each task, which is cross-entropy loss of clean input and perturbed input (Eq. 6). We employ the similarity of these gradients to assess the difficulties of optimizing architecture for both tasks. When the gradient directions are highly aligned between the two tasks, the learning trajectory for both tasks becomes more predictable, facilitating the optimization of both tasks easily. In contrast, orthogonal gradient directions lead to greater unpredictability, hindering optimization and potentially resulting in suboptimality for both clean or perturbed classification tasks. Therefore, to evaluate the stability of optimizing both tasks, we measure the absolute value of gradient similarity as follows:

$$\mathcal{G}_m(g, g^r) = \left| \frac{g_m \cdot g_m^r}{\|g_m\|\|g_m^r\|} \right|. \tag{9}$$

**Consistency-based Robust Zero-Cost Proxy: CRoZe.** In sum, to evaluate the robustness of the given architecture, we propose a scoring mechanism that evaluates the consistencies of features, parameters, and gradients between the clean network $f_\theta$ and the robust network $f_{\theta^r}$ that are obtained with a single gradient update. The robustness score for a given architecture is computed as follows:

$$\text{CRoZe}(x, x'; f_\theta, f_{\theta^r}) = \sum_{m=1}^{M} \mathcal{Z}_m \times \mathcal{P}_m \times \mathcal{G}_m. \tag{10}$$

That is, we score the network $f_\theta$ with a higher CRoZe score as more robust to perturbations. In the next section, we show that this measure is highly correlated with the actual robustness of a fully-trained model (Table 1).

## 4 Experiments

We now experimentally validate our proxy designed to identify robust architectures that perform well on both clean and perturbed inputs, on multiple benchmarks. We first evaluate Spearman's $\rho$ between robustness and the proxy's values across different tasks and perturbations in the NAS-Bench-201

Table 1: Comparison of Spearman's $\rho$ between the actual accuracies and the proxy values on CIFAR-10 in the NAS-Bench-201 search space. Clean stands for clean accuracies and robust accuracies are evaluated against diverse adversarial attacks [16, 5, 32, 38, 17, 7]. * denotes the HRS value of each attack and AA. indicates AutoAttack. Avg. stands for average Spearman's $\rho$ values with all accuracies. **Bold** and underline stands for the best and second, respectively. All models are trained on adversarially-perturbed images.

| Proxy Type | Clean | FGSM | PGD | FGSM* | PGD* | CW | DeepFool | SPSA | LGV | AA. | Avg. |
|---|---|---|---|---|---|---|---|---|---|---|---|
| FLOPs | 0.670 | 0.330 | 0.418 | 0.531 | 0.515 | 0.189 | 0.364 | 0.196 | 0.347 | 0.365 | 0.393 |
| #Params. | 0.678 | 0.341 | 0.429 | 0.541 | 0.526 | 0.182 | 0.371 | 0.209 | 0.355 | 0.375 | 0.401 |
| Plain [1] | -0.042 | -0.007 | -0.012 | -0.016 | -0.016 | 0.072 | 0.009 | 0.009 | 0.010 | 0.015 | 0.002 |
| Grasp [1] | 0.470 | 0.324 | 0.341 | 0.392 | 0.375 | 0.179 | 0.393 | **0.249** | 0.401 | 0.397 | 0.352 |
| Fisher [1] | 0.482 | 0.226 | 0.276 | 0.335 | 0.334 | 0.234 | 0.242 | 0.092 | 0.239 | 0.244 | 0.270 |
| GradNorm [1] | 0.659 | 0.336 | 0.400 | 0.490 | 0.478 | **0.264** | 0.421 | 0.149 | 0.401 | 0.405 | 0.400 |
| SynFlow [1] | 0.635 | 0.355 | 0.420 | 0.519 | 0.498 | 0.202 | 0.397 | 0.196 | 0.387 | 0.383 | 0.399 |
| NASWOT [30] | 0.600 | 0.332 | 0.381 | 0.437 | 0.438 | 0.240 | 0.250 | 0.197 | 0.265 | 0.280 | 0.342 |
| CRoZe | **0.723** | **0.417** | **0.501** | **0.602** | **0.588** | 0.220 | **0.454** | 0.240 | **0.449** | **0.458** | **0.465** |

search space, comparing it to clean zero-shot NAS methods (Section 4.2). We then evaluate the computational efficiency and final performance of the chosen architecture using our proxy in the DARTS search space, comparing it to existing robust NAS methods, which are computationally costly (Section 4.3). Lastly, we analyze the proxy's ability to accurately reflect the fully trained model's behavior in a single step, as well as the capacity of the chosen robust architecture to consistently generate features, aligned gradients, and parameters against clean and perturbed inputs (Section 4.4).

## 4.1 Experimental Setting

**Baselines.** We consider three types of existing NAS approaches as our baselines. **1) Clean one-shot NAS** [42, 6]: One-shot NAS methods for searching architectures only on clean samples. **2) Clean zero-shot NAS** [1, 30]: Zero-shot NAS with proxies that evaluate the clean performance of architectures without any training. **3) Robust NAS** [18, 31]: One-shot NAS methods for searching architectures only on adversarially perturbed samples.

**Datasets.** For the NAS-Bench-201 [12, 22] search space, we validate our proxy across different tasks (CIFAR-10, CIFAR-100, and ImageNet16-120) and perturbations (FGSM [16], PGD [29], and 15 types of common corruptions [20]). To measure Spearman's $\rho$ between final accuracies and our proxy values, we use both clean NAS-Bench-201 [12] and robust NAS-Bench-201 [22], which include clean accuracies and robust accuracies, respectively. Finally, we search for the optimal architectures with our proxy in DARTS [26] search space and compare the final accuracies against previous NAS methods [31, 18, 6, 1] on CIFAR-10 and CIFAR-100.

**Standard Training & Adversarial Training.** For a fair comparison, we use the same training and testing settings to evaluate all the architectures searched with all NAS methods, including ours. 1) Standard Training: We train the neural architectures for 200 epochs under SGD optimizer with a learning rate of 0.1 and weight decay 1e-4, and use a batch size of 64 following [31]. 2) Adversarial Training: We train the neural architectures with $l_\infty$ PGD attacks with the epsilon of 8/255, step size 2/255, and 7 steps. We evaluate the robustness against various perturbations, which are adversarial attacks (FGSM [16], PGD [29], CW [5], DeepFool [32], SPSA [38], LGV [17], and AutoAttack [7]) and common corruptions [20]. More experimental details are described in Supplementary A.

## 4.2 Results on NAS-Bench-201

**Standard-Trained Neural Architectures.** We verify our proxy's capability to search for high-performing neural architectures across various tasks and perturbations by employing Spearman's $\rho$ to measure the rank correlation between the proxy values and final accuracies [1, 30, 11]. While existing clean zero-shot NAS works [1, 30] display weaker correlations with clean accuracies than using the number of parameters (#Params.) as a proxy, our proxy exhibits significantly stronger correlations with clean accuracies across all tasks. Specifically, it shows improvements of 5.92% and 1.95% on CIFAR-10 and CIFAR-100, respectively, compared to best-performing baselines (Table 2).

Furthermore, CRoZe shows remarkable Spearman's $\rho$ for robust accuracies obtained against adversarial perturbations and corrupted noises across tasks. Notably, our proxy outperforms the SynFlow [1] by 6.41% and 8.77% in an average of Spearman's $\rho$ for adversarial perturbations and common cor-

Table 2: Comparison of Spearman's $\rho$ between the actual accuracies and the proxy values on CIFAR-10, CIFAR-100, and ImageNet16-120 in NAS-Bench-201 search space. Clean stands for clean accuracies and robust accuracies are evaluated against adversarial perturbations (FGSM) [16] with various attack sizes ($\epsilon$) and common corruptions [20]. Avg. stands for average Spearman's $\rho$ values with all accuracies within each task. **Bold** and underline stands for the best and second, respectively. All models are standard-trained on clean images.

| Proxy Type | CIFAR-10 | | | | | | | | CIFAR-100 | | | | | | | ImageNet16-120 | | |
| | Clean | FGSM | | Common Corruption (CC.) | | | | Avg. | Clean | FGSM | Common Corruption (CC.) | | | | Avg. | Clean | FGSM | Avg. |
| | | $\epsilon=8$ | $\epsilon=4$ | Weather | Noise | Blur | Digital | | | $\epsilon=4$ | Weather | Noise | Blur | Digital | | | $\epsilon=4$ | |
|---|---|---|---|---|---|---|---|---|---|---|---|---|---|---|---|---|---|---|
| FLOPs | 0.726 | 0.753 | 0.740 | 0.665 | 0.138 | 0.232 | 0.473 | 0.532 | 0.699 | 0.661 | 0.674 | 0.253 | 0.492 | 0.607 | 0.557 | 0.680 | 0.611 | 0.634 |
| #Params. | 0.747 | 0.756 | 0.739 | 0.674 | 0.131 | 0.229 | 0.489 | 0.502 | 0.720 | 0.654 | 0.685 | 0.240 | 0.489 | 0.618 | 0.568 | 0.683 | 0.627 | 0.655 |
| Plain [1] | -0.073 | -0.059 | -0.055 | -0.041 | 0.035 | 0.048 | -0.032 | -0.025 | -0.242 | -0.172 | -0.169 | 0.055 | -0.110 | -0.182 | -0.137 | -0.231 | -0.241 | -0.236 |
| Grasp [1] | 0.551 | 0.657 | 0.673 | 0.641 | **0.250** | 0.226 | 0.426 | 0.532 | 0.550 | 0.649 | 0.565 | 0.311 | 0.468 | 0.518 | 0.510 | 0.543 | 0.601 | 0.572 |
| Fisher [1] | 0.383 | 0.466 | 0.499 | 0.411 | 0.179 | 0.221 | 0.244 | 0.343 | 0.382 | 0.575 | 0.425 | 0.325 | 0.447 | 0.396 | 0.425 | 0.324 | 0.394 | 0.359 |
| GradNorm [1] | 0.637 | 0.760 | 0.774 | 0.540 | 0.239 | 0.183 | 0.375 | 0.460 | 0.638 | **0.789** | 0.654 | **0.357** | **0.562** | 0.613 | 0.602 | 0.573 | 0.649 | 0.611 |
| SynFlow [1] | 0.777 | 0.778 | 0.751 | 0.673 | 0.188 | 0.181 | 0.554 | 0.557 | 0.769 | 0.683 | 0.703 | 0.215 | 0.439 | 0.641 | 0.575 | 0.751 | 0.695 | 0.723 |
| NASWOT [30] | 0.687 | 0.522 | 0.513 | 0.479 | -0.078 | 0.012 | 0.434 | 0.367 | 0.708 | 0.520 | 0.525 | -0.023 | 0.319 | 0.534 | 0.431 | 0.698 | **0.730** | 0.714 |
| CRoZe | **0.823** | **0.826** | **0.801** | **0.743** | 0.190 | **0.236** | **0.565** | **0.598** | **0.784** | 0.693 | **0.747** | 0.251 | 0.504 | **0.682** | **0.610** | **0.765** | 0.696 | **0.731** |

(a) Search on Clean Inputs      (b) Search on Adversarially Perturbed Inputs

Figure 3: **Search with our proxy on CIFAR-10 in NAS-Bench-201 search space.** Our proxy can reduce the sampling costs in searching for neural architectures against both clean (left) and perturbed samples (right).

ruptions on CIFAR-10, respectively (Table 2). Our results on multiple benchmark tasks with diverse perturbations highlight the ability of our proxy to effectively search for robust architectures that can consistently outperform predictions against various perturbations. Importantly, our proxy is designed to prioritize generalizability, and as a result, it exhibits consistently enhanced correlation with final accuracies for both clean and perturbed samples. This result indicates that considering generalization ability is effective in identifying robust neural architectures against diverse perturbations but also leads to improved performance for clean neural architectures.

**Adversarially-Trained Neural Architectures.** We also validate the ability of our proxy to precisely predict the robustness of adversarially-trained networks, specifically for adversarial perturbations. Adversarial training [29] is a straightforward approach to achieving robustness in the presence of adversarial perturbations. To assess the Spearman's $\rho$ of robustness in adversarially-trained networks, we construct a dataset consisting of final robust accuracies of 500 randomly sampled neural architectures from NAS-Bench-201 search space that are adversarially-trained [29] from scratch. The final robust accuracies are obtained for 7 different adversarial attacks, including FGSM, PGD, CW, DeepFool, SPSA, LGV, and AutoAttack.

Notably, our proxy showcases the highest overall correlation for the 7 different adversarial attacks, supporting the ability of our proxy to search the architectures that are robust against diverse perturbations. Considering the trade-off between the clean and robust accuracy in adversarial training [43], we employ the harmonic robustness score (HRS) [9] to evaluate the overall performance of the adversarially-trained models. When comparing the correlations between clean performances and proxy values, existing clean zero-shot NAS methods, i.e., SynFlow and Grasp, demonstrate higher correlations in the FGSM or PGD, but their correlations with clean accuracies are poorer than Grad-Norm and Fisher, respectively. This result shows that clean zero-shot NAS methods tend to search for architectures that are more prone to overfitting to either clean or robust tasks (Table 1, 2). In contrast, CRoZe consistently achieves higher Spearman's $\rho$ for both clean and robust tasks, ultimately enabling the search for architecture with high HRS due to consideration of alignment in gradients.

## 4.3 End-to-End Generalization Performance on DARTS

In this section, we evaluate the effectiveness of CRoZe in rapidly searching for generalized neural architectures in the DARTS search space and compare it with previous clean one-shot NAS [42, 6], clean zero-shot NAS [37], and robust NAS [31, 18] in terms of performance and computational cost.

**Efficient Sampling with CRoZe.** The efficiency of our proxy lies in evaluating neural architectures without iteratively training models. As a result, most search time is dedicated to sampling the

Table 3: Comparisons of the final performance and search time of the searched network in DARTS search space on CIFAR-10 and CIFAR-100. All NAS methods are executed on a single NVIDIA 3090 RTX GPU.

| Task | NAS Method | Training-free NAS | Params (M) | Search Time (GPU sec) | Clean | CC. | FGSM | HRS |
|------|-----------|-------------------|------------|----------------------|-------|-----|------|-----|
| | | | | | \multicolumn{4}{c}{Standard-Trained} | | | |
| CIFAR-10 | PC-DARTS [42] | | 3.60 | 8355 | **95.14** ($\pm$ 0.16) | 73.57 ($\pm$ 0.89) | 17.74 ($\pm$ 3.44) | 29.76 ($\pm$ 4.81) |
| | DrNAS [6] | | 4.10 | 46857 | 94.18 ($\pm$ 0.33) | 71.79 ($\pm$ 0.74) | 15.63 ($\pm$ 2.78) | 26.71 ($\pm$ 4.01) |
| | RobNet [18] | | 5.44 | 274062 | 94.94 ($\pm$ 0.26) | 72.49 ($\pm$ 0.45) | 14.69 ($\pm$ 1.15) | 25.42 ($\pm$ 1.71) |
| | AdvRush [31] | | 4.20 | 251245 | 93.44 ($\pm$ 0.41) | 71.43 ($\pm$ 1.07) | 14.94 ($\pm$ 2.56) | 25.67 ($\pm$ 3.87) |
| | GradNorm [1] | ✓ | 4.69 | 9740 | 92.95 ($\pm$ 0.67) | 63.03 ($\pm$ 6.33) | 15.64 ($\pm$ 0.15) | 26.78 ($\pm$ 0.24) |
| | SynFlow [1] | ✓ | 5.08 | 10138 | 92.15 ($\pm$ 1.25) | 70.74 ($\pm$ 2.69) | 12.65 ($\pm$ 1.46) | 22.22 ($\pm$ 2.32) |
| | CRoZe | ✓ | 5.52 | 17066 | 94.34 ($\pm$ 0.08) | **74.11** ($\pm$ 1.40) | **20.51** ($\pm$ 1.32) | **33.07** ($\pm$ 2.30) |
| CIFAR-100 | PC-DARTS [42] | | 3.60 | 8355 | **76.95** ($\pm$ 0.67) | **50.02** ($\pm$ 0.05) | 7.68 ($\pm$ 0.47) | 13.96 ($\pm$ 0.80) |
| | DrNAS [6] | | 4.10 | 46857 | 75.49 ($\pm$ 1.66) | 49.18 ($\pm$ 1.31) | 8.14 ($\pm$ 0.22) | 14.69 ($\pm$ 0.35) |
| | RobNet [18] | | 5.44 | 274062 | 76.40 ($\pm$ 0.30) | 49.77 ($\pm$ 0.82) | 7.50 ($\pm$ 0.84) | 13.65 ($\pm$ 1.39) |
| | AdvRush [31] | | 4.20 | 251245 | 74.48 ($\pm$ 1.35) | 48.27 ($\pm$ 1.42) | 7.46 ($\pm$ 0.81) | 13.55 ($\pm$ 1.37) |
| | GradNorm [1] | ✓ | 3.83 | 9554 | 68.52 ($\pm$ 0.80) | 43.69 ($\pm$ 2.24) | 6.87 ($\pm$ 1.58) | 12.44 ($\pm$ 2.61) |
| | SynFlow [1] | ✓ | 4.42 | 9776 | 76.23 ($\pm$ 0.73) | 49.60 ($\pm$ 0.78) | 9.01 ($\pm$ 0.56) | 16.11 ($\pm$ 0.91) |
| | CRoZe | ✓ | 4.72 | 17457 | 75.46 ($\pm$ 0.81) | 49.33 ($\pm$ 0.38) | **9.75** ($\pm$ 0.78) | **17.26** ($\pm$ 1.22) |

candidate architectures. To thoroughly explore the sampling costs involved in identifying good neural architectures, we conduct experiments in the NAS-Bench-201 search space with two representative sampling-based search algorithms: random search (RAND) and evolutionary search (AE).

Our experiments show that CRoZe can rapidly identify an architecture with 91.5% clean accuracy and 50% FGSM robustness, even early in the search. By leveraging the initial pool of architectures from CRoZe with high proxy values (RAND+warmup and AE+warmup), our proxy cuts sampling costs and rapidly identifies high-performing architectures for clean inputs (Figure 3a). Moreover, by focusing sampling around the architecture with the highest proxy values of CRoZe (AE+warmup+move), we can find even more robust architectures using fewer sampled models ($< 50$) (Figure 3b). In essence, CRoZe effectively identifies high-performing neural architectures for both clean and perturbed inputs in the NAS-Bench-201 search space.

**End-to-End Performance.** We validate the final performance of the neural architectures discovered by CRoZe (Table 3, 4) and compare the search time and performance with existing NAS methods including robust one-shot NAS (RobNet [18] and AdvRush [31]), clean one-shot NAS (PC-DARTS [42] and DrNAS [6]), and clean zero-shot NAS (SynFlow, GradNorm [1]). For a fair comparison between clean zero-shot NAS (SynFlow, GradNorm) and CRoZe, we sample the same number of candidate architectures (e.g., 5,000) using the warmup+move strategy in the DARTS search space. We report the average and standard deviation of accuracies evaluated over 3 different neural architecture searches. All experiments are conducted on a single NVIDIA 3090 RTX GPU to measure search costs.

In standard training, the searched network by our proxy surpasses the networks by robust one-shot NAS methods, RobNet and AdvRush, in terms of robust accuracy against FGSM on CIFAR-10, improving by 5.82% and 5.57%, respectively (Table 3). CRoZe also shows the highest HRS accuracy, 7.40% better than AdvRush on CIFAR-10, indicating its efficiency in mitigating the trade-off between clean and robust accuracy, while reducing the search cost by 14.7 times.

Table 4: Final performance of the searched networks that are adversarially-trained on CIFAR-10.

| NAS Method | Clean | Robustness | | | | | |
|-----------|-------|-----|-----|------|-----|-----|------|
| | | PGD | CW | SPSA | LGV | AA. | Avg. |
| PC-DARTS | 86.02 | 52.16 | 9.85 | 86.00 | 79.32 | 48.82 | 55.23 |
| DrNAS | 86.45 | 54.66 | 6.44 | 85.86 | 79.64 | 52.40 | 55.80 |
| RobNet | 80.53 | 50.62 | 22.91 | 80.85 | 77.79 | 46.34 | 55.70 |
| AdvRush | 85.98 | 53.89 | 6.68 | 84.81 | 79.61 | 51.88 | 55.37 |
| GradNorm | 81.61 | 49.86 | 12.02 | 77.18 | 73.27 | 46.69 | 51.61 |
| SynFlow | 77.08 | 45.95 | 26.50 | 75.78 | 74.14 | 42.45 | 52.96 |
| CRoZe | 84.28 | 52.17 | 19.13 | 83.43 | 76.85 | 48.14 | **55.94** |

Compared to the previously best-performing clean zero-cost proxy, SynFlow, CRoZe finds architectures with superior performance in clean, common corruptions, and FGSM scenarios, showcasing the effectiveness of our proxy in identifying generalized architectures. Moreover, the architecture chosen by CRoZe outperforms clean one-shot NAS methods in HRS accuracy, addressing vulnerability against both perturbations and clean inputs effectively (Figure 2a). When we adversarially train the searched architectures, CRoZe achieves the highest average robustness against 5 adversarial attacks at much lower search costs compared to robust NAS on CIFAR-10 (Table 4, Figure 2b).

## 4.4 Further Analysis

**Ablation of Each Component of CRoZe.** Our proxy consists of three components: feature, parameter, and gradient consistency. To analyze the importance of considering all factors for evaluating the robustness of the given neural architecture, we conduct an ablation study in both the

Table 5: Comparisons of the final performance of the searched network in NAS-Bench-201 and DARTS search space on CIFAR-10. All models are standard-trained. **Bold** and underline stands for the best and second.

| Proxy components | | | NAS-Bench-201 | | | | | DARTS | | | | |
|---|---|---|---|---|---|---|---|---|---|---|---|---|
| Feature | Parameter | Gradient | Clean | CC. | FGSM | HRS | Avg. | Clean | CC. | FGSM | HRS | Avg. |
| ✓ | – | – | 93.30 | 55.62 | 44.30 | 60.08 | 63.33 | 94.37 | 72.26 | 16.87 | 28.62 | 53.03 |
| ✓ | ✓ | – | 93.70 | 56.93 | 45.80 | 61.53 | 64.49 | **94.99** | 74.06 | 16.82 | 28.58 | 53.61 |
| ✓ | – | ✓ | 93.70 | 56.93 | 45.80 | 61.53 | 64.49 | 94.30 | **74.91** | 16.67 | 28.33 | 53.55 |
| – | ✓ | ✓ | 93.70 | 56.93 | 45.80 | 61.53 | 64.49 | 94.34 | 74.46 | 15.71 | 26.93 | 52.86 |
| ✓ | ✓ | ✓ | 93.70 | 56.93 | 45.80 | 61.53 | 64.49 | 94.45 | 74.63 | **22.38** | **36.19** | **56.91** |

Figure 5: Spearman's $\rho$ in our proxy and each consistency component between the architectures with single-step trained states and two of fully-trained and 10-step trained states, respectively, against clean and perturbed images.

NAS-Bench-201 search space and the DARTS search space. On the NAS-Bench-201 search space, all ablation proxies identified the same candidate architecture except for the feature-based proxy. To compare the ablation proxies thoroughly, we extend our examination to the DARTS search space, which contains a significantly larger number of candidate architectures (e.g., $10^{19}$) compared to the NAS-Bench-201 search space. As shown in Table 5, architecture identified solely by feature consistency exhibits better average performance compared to those discerned by proxies without feature consistency, emphasizing the influential role of feature consistency in evaluating robustness. Notably, the architecture selected by our proxy showcases the highest average performance. This indicates that both parameter and gradient consistency can bolster feature consistency, offering more insight into the learning trajectory, as elaborated in Section 3.3. Overall, considering all components is crucial to evaluate the robustness within a single batch.

**Feature Variance of the Robust Models.** Our proxy searches for architectures that can generalize to all types of perturbed inputs, which are not over-fitted to a single type of perturbation, as described in Section 3.1 (Eq. 2). To validate this, we analyzed the feature variance of neural architectures that demonstrated the highest performance against a single type of perturbation, namely FGSM and PGD, compared

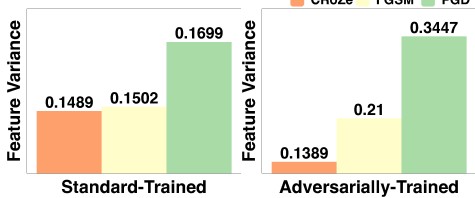

Figure 4: Feature variance across perturbations.

to an architecture selected with the CRoZe proxy from a pool of 300 standard-trained and adversarially-trained models (Figure 4). Features were obtained from 18 different perturbations, including clean, PGD, FGSM, and 15 types of common corruptions. Remarkably, the neural architectures selected with our proxy exhibited the smallest standard deviations between the features obtained from the 18 different perturbations, in both standard and adversarially-trained scenarios. Conversely, the architecture with the best PGD robustness demonstrated 2.48 times larger variations in adversarially-trained scenarios, indicating a higher risk of overfitting to specific perturbations. These results provide compelling evidence of the effectiveness of CRoZe in identifying robust architectures that can consistently extract features and generalize across diverse perturbations.

**Assessment of CRoZe Predictiveness.** To empirically validate whether our proxy obtained from a random state accurately represents the proxy in the fully-trained model, we conducted an analysis using Spearman's $\rho$. We randomly sampled 300 architectures from the NAS-Bench-201 search space and trained them on the entire dataset, using both the full training and reduced training only of 10 steps. The Spearman's $\rho$ between the proxy value obtained from a single-step trained state and the 10-step trained state shows a strong correlation of 0.995. Even after full training, the correlation remained high at 0.970 (Figure 5). These suggest that our proxy, derived from estimated surrogate networks, can significantly reduce the computational costs associated with obtaining fully-trained models. Furthermore, each component of CRoZe consistently demonstrates a high correlation with

the fully-trained states, supporting the notion that the high correlation computed with the final state is not solely a result of the collective influence of components, but rather an accurate estimation.

**Validation of CRoZe for Estimating Fully-Trained Neural Architectures.** To investigate whether the architectures selected with our proxy possess the desired robustness properties discussed in Section 3.1, we conduct an analysis. From a pool of 300 samples in the NAS-Bench-201 search space, we select top-5 and bottom-5 architectures based on our proxy, excluding those with a robust accuracy of less than 20%. We evaluate the performances of these architectures that are standard-trained (ST) and adversarially-trained (AT) against clean inputs and three distinct types of perturbed inputs: adversarial perturbations (i.e., FGSM and PGD), and common corruptions.

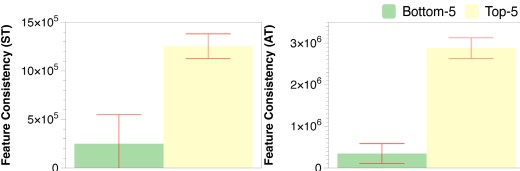

Figure 6: Comparisons of feature consistency in fully-trained states.

Notably, the top-5 networks show impressive average clean accuracies of 89.62%, and 84.36% on CIFAR-10, while the bottom-5 networks achieve significantly lower average clean accuracies of 61.34%, 51.64% in standard training and adversarial training, respectively (Table 6). Moreover, the top-5 networks exhibit an average robust accuracy of 54.95% against perturbations, compared to only 28.50% for the bottom-5 group. Furthermore, by considering feature consistency between clean and perturbed inputs, as defined in Eq. 2, the top-5 group exhibited similar features, whereas the bottom-5 group showed dissimilar features (Figure 6). This supports that architectures with consistent features across inputs are more likely to make accurate predictions under perturbations. Overall, our analysis provides strong evidence that our proxy can search for generalizable robust architectures that accurately predict the final accuracies under diverse perturbations.

|  | Standard-Trained (ST) | | | | |
|---|---|---|---|---|---|
| Type | Clean | CC. | PGD | FGSM | Avg. Rob. |
| Top-5 | 89.62 | 77.68 | 26.09 | 61.08 | 54.95 |
| Bottom-5 | 61.34 | 49.13 | 19.35 | 17.01 | 28.50 |
|  | Adversarially-Trained (AT) | | | | |
|  | Clean | PGD | HRS(PGD) | FGSM | HRS(FGSM) |
| Top-5 | 84.36 | 40.55 | 54.76 | 66.48 | 74.34 |
| Bottom-5 | 51.64 | 22.59 | 31.36 | 37.20 | 43.04 |

Table 6: Comparisons of the performance.

## 5 Conclusion

While neural architecture search (NAS) is a powerful technique for automatically discovering high-performing deep learning models, previous NAS works suffer from two major drawbacks: computational inefficiency and compromised robustness against diverse perturbations, which hinder their applications in real-world scenarios with safety-critical applications. In this paper, we proposed a simple yet effective lightweight robust NAS framework that can rapidly search for well-generalized neural architectures against diverse tasks and perturbations. To this end, we proposed a novel consistency-based zero-cost proxy that evaluates the robustness of randomly initialized neural networks by measuring the consistency in their features, parameters, and gradients for both clean and perturbed inputs. Experimental results demonstrate the effectiveness of our approach in discovering well-generalized architectures across diverse search spaces, multiple datasets, and various types of perturbations, outperforming the baselines with significantly reduced search costs. Such simplicity and effectiveness of our approach open up new possibilities for automatically discovering high-performing models that are well-suited for safety-critical applications.

## Acknowledgement

This work was supported by the Institute of Information & communications Technology Planning & Evaluation (IITP) grant funded by the Korea government (MSIT) (No.2020-0-00153), the National Research Foundation of Korea(NRF) grant funded by the Korea government(MSIT) (No. RS-2023-00256259), the Institute of Information & communications Technology Planning & Evaluation (IITP) grant funded by the Korea government(MSIT) (No.2019-0-00075, Artificial Intelligence Graduate School Program(KAIST)), Center for Applied Research in Artificial Intelligence (CARAI) grant funded by DAPA and ADD (UD190031RD), and Google Research Grant and Google Cloud Research Credits program with the award (WLKY-E43Y-19JJ-A0E2). We thank Jin Myung Kwak, Eunji Ko, Yulmu Kim, Sohyun An, and Hayeon Lee for providing helpful feedback and suggestions in preparing an earlier version of the manuscript. We also thank the anonymous reviewers for their insightful comments and suggestions.

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
