# OpenReview forum: "Generalizable Lightweight Proxy for Robust NAS against Diverse Perturbations"
_NeurIPS.cc/2023/Conference — NeurIPS 2023 poster_

### Official Review · Reviewer_KEHv · 2023-06-18

**Soundness:** 3 good
**Presentation:** 2 fair
**Contribution:** 2 fair
**Rating:** 4
**Confidence:** 4

**Summary:**

The paper concerns the automatic generation of architectures that are robust to diverse perturbations. Neural architecture search (NAS) has been used for the automatic generation of such architectures, but the paper notes that most of those architectures are dedicated to the clean accuracy, which leaves the resulting architectures unprotected against (adversarial) perturbations. One of the reasons that the previous works have focused on clean accuracy is that making the architecture robust to perturbations is costlier and results in methods that are computationally heavy. This is precisely the gap the paper intends to fill by proposing a lightweight way to obtain robust architectures. Concretely, it considers a "zero-cost proxy" that considers the consistency across features, parameters and gradients of clean/perturbed images. The exact heuristic, called croze, is provided in eq. 10 and accounts for the aforementioned consistency across features, parameters and gradients. The method is empirically validated in small scale datasets of cifar10, cifar100 and imagenet16.


**Update**: I am thankful to the authors for their answers. I strongly believe that the discussion and points below should be included in the camera-ready version, along with the results on accuracy. Practitioners care about the accuracy in both clean and robust accuracy and this is the metric that should be compared with standard benchmarks. In addition, the insights reported during the rebuttal period might be interesting to the reader, therefore I would urge the authors to include the relevant insights in the paper.

**Strengths:**

NAS is an important way to discover new architectures and as such I believe the paper is relevant to the community. In addition, the paper proposes a method to obtain robust architectures, which is a topic of intense research in the ML community. In addition, I find the figures 1,2 quite clear in the message they want to convey. Having said that, the paper raises a lot of questions and could be improved (see below).

**Weaknesses:**

The paper is not entirely clear in a number of paragraphs and legends. For instance, in fig. 2 the CC Accuracy and HRS accuracy are not referred to in the introduction, so they leave the reader searching or wondering about them. In addition, the related work with respect to train-free NAS is slightly outdated and more methods do mitigate the issue the paper raises, i.e. the computational cost. For instance, the papers [1-3] are directly relevant. It is also recommended to include those in the related experimental validation.

What is more, I find the experimental validation slightly unusual, since typically in recognition we are interested in the clean/robust accuracy, especially when comparing with other methods.

Minor: The paper would benefit from proofreading since there are several mistakes or phrases that are unclear. For instance, "the gradients similarity as an evaluating", "poorer correlations" (line 263), "using a less number of sampled words".


[1] NASI: Label- and Data-agnostic Neural Architecture Search at Initialization, ICLR’22.

[2] Knas: Green neural architecture search, ICML’21.

[3] Generalization Properties of NAS under Activation and Skip Connection Search, NeurIPS’22.

**Questions:**

The paper mentions that previous NAS approaches paid less attention to robustness. "This can result in [...] limiting the practical deployment of NAS in real-world safety-critical applications". There are no citations or references about such safety-critical applications for NAS, so I am wondering what those applications are.

The paper claims that "to have a correct prediction on the perturbed input x', the model needs to extract similar features between x' and x". Is there any proof that this is a necessary condition? Otherwise, I find this a strong claim that is nontrivial to me. In addition, I am wondering whether eq. 2 is a sufficient condition and whether there is a reference for this, or what the norm is.

In sec. 3.3 there are a number of unclear claims to me. Firstly, why should the features from two different functions $f_{\theta}$ and $f_{\theta^r}$ be similar? Secondly, what is the $z_m$ in eq. 5? Is it the output of $e_{\psi}$ from sec. 3.1? Thirdly, in high-dimensions, aren't the features very likely to be (near) orthogonal? So, I am not sure what the eq. 5 can capture in realistic networks that have representations of thousands or millions dimensions.

The paper mentions that "Accordingly, since the higher similarity of single-step updated parameters may promote the model to converge to an identical or similar parameter space for both tasks, we evaluate the parameter similarity as one of our proxy terms as follows:". I am wondering what the last part, i.e. proxy term, means in this context. How does this proxy term mitigate the drawback mentioned in the previous sentence?

As I understand, the final cost is the one in eq. 10, but I am wondering why the multiplication is selected for this. Is there any intuition for this?

The reference [17] mentioned as a benchmark is only evaluating the robustness at test-time, right? The architectures are trained with clean training.

What is the 0.747 in the number of parameters (e.g. in table 1)? Is this million parameters? This is not clear to me.

**Limitations:**

The limitations are mentioned in the supplementary. The text "while existing NAS frameworks require enlarged computational resources when utilizing larger datasets such as ImageNet, our method requires constant computational resources that are unaffected by the specific tasks" mentions ImageNet, but this paper does not do search on ImageNet.

---

> ### Author Rebuttal · Authors · 2023-08-07
>
> **W1. A number of paragraphs and legends are not clear.**
> * We will clarify that CC refers to the Common Corruption dataset, and HRS accuracy stands for Harmonic Robust Score, which is the harmonic average of clean and robust accuracy in the paper, and change the names accordingly.
> ---
> **W2. Prior train-free NAS works are outdated. NASI, Knas, and Eigen-NAS should be included.**
> * We will include NASI, KNAS, and Eigen-NAS in the related work. Moreover, **in Table R5 of the PDF, we verified the effectiveness of our proxy** in identifying the robust architecture compared to those works. CRoZe **exhibits consistently the highest clean and robust accuracies** on CIFAR10 and CIFAR100 compared to recent train-free NAS works, **outperforming the best baseline (KNAS) by 0.32% and 4.26%** for clean and average robust accuracies on CIFAR10.
> ---
> **W3. Experimental validation seems unusual.**
> * Our approach aims to seek an architecture that performs well on both clean and perturbed inputs. Thus, we validate our proxy by measuring the correlation between our proxy score at initialization and the final performance of architectures from the search space. **This follows the protocol used in prior works on zero-cost proxy[1,2]**.
> * **In Table 3, we present results that demonstrate the superiority of our proxy in terms of clean and robust accuracy, following an end-to-end evaluation similar to [3]**. Moreover, in **Table R2 of PDF**, we show end-to-end NAS results in adversarial training settings, achieving the **highest average robust accuracy of  55.77% with 15 times less search cost** compared to AdvRush. From these experimental validations, we believe our proxy is able to find robust architecture rapidly.
> ---
> **W4. Paper needs proofreading.**
> * We will incorporate the corrections in the revised version.
>    - line 188: Thus, we employ gradient similarity as a means of evaluating the ~.
>    - line 263: lower correlations
> ---
> **Q1. No citations on safety-critical applications for NAS.**
> * NAS[4,5] is used to find the optimal architecture for various tasks of real-world applications, which may be safety-critical, such as autonomous driving systems [6]. If NAS does not consider robustness in such a setting, the resulting system may be vulnerable to perturbations.
> ---
> **Q2. Is there evidence that the model needs to extract similar features against clean and perturbed images for correct prediction?**
> * Our proxy is designed on top of the theoretical understanding that robust architectures learn invariant features against clean and perturbed images [7,8,9], irrespective of type of the perturbations applied.
> * We assume that 1) the neural network is continuous and 2) semantic-preserving perturbations satisfying $||x^{’}-x||<\delta$ hold true, where $\delta$ is sufficiently small bound. We will clarify that $||.||$ is the $L_p$ norm.
> ---
> **Q3. Why should features from two different functions be similar? Is eq.5 the output from sec 3.1? Does eq.5 capture realistic networks with high dimensions?**
> * Since the two functions represent the clean surrogate network (randomly initialized) and the robust surrogate network (perturbed weights), both within the same architecture, we can calculate the feature difference between the clean and robust model. This proxy term is inspired by how robust models extract perturbation-invariant features from clean and robust inputs.
> * $z_m$ is the output vector of the $m^{th}$ layer of $f_\theta$. We will clarify this in the manuscript.
> * Even in high dimensions, Eq.5 can capture whether the model can extract invariant features from clean and perturbed inputs. Specifically, features from clean and perturbed images with a dimension of 512 show a high similarity of 0.983, which is not orthogonal.
> ---
> **Q4. Exact context of lines 183-185.**
> * The proxy term is used to evaluate the parameter similarity to estimate how the model will converge to an identical or similar parameter space for both tasks. By measuring the similarity from the single-step updated parameters, we can assess how well the convergence process is likely to proceed.
> ---
> **Q5. Why the multiplication is selected for the proxy?**
> * As shown in Figure 1 (b), we believe each component should be proportional to the final cost in eq.10. Therefore, we employ the multiplication of each component for the final proxy.
> ---
> **Q6. The referenced benchmark evaluates the robustness only at test-time?**
> * Yes. To provide a more comprehensive robust evaluation, we further provide results (Right, Table 1) with robustness against diverse perturbations of adversarially-trained architectures from the subset of the NAS-Bench-201 search space.
> ---
> **Q7. Meaning of 0.747 in table 1.**
> * The value 0.747 represents **Spearman’s rank correlation coefficient value** between the rank derived from the number of parameters and the rank derived from the final validation accuracies.
> ---
> **L1. The paper mentions ImageNet but no experiments on that.**
> * **We provide the experimental results on ImageNet 16-120 in Table 2**. ImageNet16-120 is a widely used benchmark in NAS works [1,2,10], which is a downsample variant of ImageNet with 151.7K training instances for 120 selected classes. We will update the paper to explicitly refer to ImageNet16-120 in the revised version to ensure clarity.
> ---
> [1]Zero-Cost Proxies for Lightweight NAS\
> [2]Neural Architecture Design and Robustness: A Dataset\
> [3]Meta-prediction Model for Distillation-aware NAS on Unseen Datasets\
> [4]PC-DARTS: Partial Channel Connections for Memory-Efficient Architecture Search\
> [5]Once-for-All: Train One Network and Specialize it for Efficient Deployment\
> [6]Robust Physical-World Attacks on Deep Learning Visual Classification\
> [7]Feature Denoising for Improving Adversarial Robustness\
> [8]Adversarial Examples Are Not Bugs, They Are Features\
> [9]Explaining and Harnessing Adversarial Examples\
> [10]NAS-Bench-201: Extending the Scope of Reproducible Neural Architecture Search

---

> > ### Comment · Reviewer_KEHv · 2023-08-14
> > **Response to the authors**
> >
> > Dear authors,
> >
> > I am thankful for your responses. However, I still have questions that are not addressed from the rebuttal:
> >
> > * The references [4, 5] mentioned above do not address the claim "This can result in [...] limiting the practical deployment of NAS in real-world safety-critical applications". Are there any references for this?
> >
> > * It's still not clear to me why the main tables are focusing on correlations other than clean/robust accuracy. I notice the same applies above to some of the responses. Could the authors elaborate on that?
> >
> > * In addition, this response is unclear to me: "To provide a more comprehensive robust evaluation, we further provide results (Right, Table 1) with robustness against diverse perturbations of adversarially-trained architectures from the subset of the NAS-Bench-201 search space". Are there any details on the paper about this subset? How was this selected?
> >
> > * Are there any insights from the components selected by NAS?

---

> > > ### Author Response · Authors · 2023-08-14
> > >
> > > Dear Reviewer,
> > >
> > > We deeply appreciate the time and effort you've dedicated to reviewing our paper. During the remaining discussion period, we will do our best to address your concerns.
> > >
> > > We provide detailed responses for your additional concerns in the below. If you have any further concerns about our work, do not hesitate to share your comments. We would be delighted to address any additional questions or concerns you may have.
> > >
> > > Best,\
> > > Authors
> > >
> > > ---
> > >
> > > **Q1. Are there any citations on safety-critical applications for NAS?** \
> > > For clarification, we elaborate our motivations, and necessity of robust NAS with the detailed references.
> > > NAS [4,5] searches for high-performing neural architectures tailored for specific tasks or datasets within a constrained computational budget. Thus, NAS can be applied to a broad range of applications, including safety-critical ones such as object detection [2] of autonomous driving system [1], image recognition [3] of manufacturer system [6], mobile systems [7], speech recognition system [8] or medical imaging systems [9]. However, the need for robust NAS [10,11,12] has recently emerged, and prior works often overlooked the importance of robustness during neural architecture searches. Consequently, the vulnerability of neural architectures discovered by previous NAS methods is inevitable [13,14] when faced with diverse perturbations [15,16,17].
> > >
> > >
> > > [1] Autonomous Driving with Deep Learning: A Survey of State-of-Art Technologies\
> > > [2] Progressive differentiable architecture search: Bridging the depth gap between search and evaluation.\
> > > [3] Neural architecture search with reinforcement learning.\
> > > [4] PC-DARTS: Partial Channel Connections for Memory-Efficient Architecture Search\
> > > [5] Once-for-All: Train One Network and Specialize it for Efficient Deployment\
> > > [6] Using Deep Learning to Detect Defects in Manufacturing: A Comprehensive Survey and Current Challenges\
> > > [7] Mnasnet: Platform-aware neural architecture search for mobile.\
> > > [8] Nas-bench-301 and the case for surrogate benchmarks for neural architecture search.\
> > > [9] NAS-Unet: Neural architecture search for medical image segmentation.\
> > > [10] Neural Architecture Design and Robustness: A Dataset\
> > > [11] Neural Architecture Search: A Survey\
> > > [12] Neural Architecture Search: Insights from 1000 Papers\
> > > [13] On the security risks of automl\
> > > [14] AdvRush: Searching for Adversarially Robust Neural Architectures\
> > > [15] Benchmarking Neural Network Robustness to Common Corruptions and Perturbations\
> > > [16] Towards Deep Learning Models Resistant to Adversarial Attacks\
> > > [17] Robust Physical-World Attacks on Deep Learning Visual Classification\
> > >
> > > ---
> > >
> > > **Q2 Could the authors elaborate on why the main tables are focusing on correlation?**\
> > > To address your concerns, **we will modify the main table with Table 3 which shows the final clean/robust accuracy** of the architecture searched by our proxy in the DARTS search space. Additionally, we provide **experimental results in the NAS-Bench-201 search space with final clean/robust accuracy in Table R5 of the PDF file** (Summarized in the following Table).
> > > However, we kindly ask you to note that **rank correlation is also an important metric, widely used in train-free NAS studies [1,2,3,4]**. The primary goal of the zero-cost proxy is to rapidly and accurately estimate the final performance of a neural architecture at the initialization state. Therefore, the rank correlation between the final performance and the value of the proxy effectively demonstrates how well our proxy is designed to search for high-performing architectures and its applicability to diverse NAS tasks.
> > >
> > > **[NAS-Bench-201 search space]**
> > > | |Standard-Trained| | | |
> > > |-|-|-|-|-|
> > > |Proxy|Clean|FGSM|PGD|CC.|
> > > |NASWOT|92.96|59.90|41.70|35.19|
> > > |NASI(T)|93.08|62.60|41.10|34.99|
> > > |NASI(4T)|93.55|64.90|44.00|36.12|
> > > |Eigen-NAS|93.46|59.60|36.80|36.75|
> > > |KNAS|93.38|63.80|44.90|34.54|
> > > |CRoZe|**93.70**|**68.00**|**48.20**|**38.83**|
> > >
> > > **[DARTS search space]**
> > > | | |Standard-Trained| |Adversarially-Trained| | |
> > > |-|-|-|-|-|-|-|
> > > |NAS Type|Method|Clean|Rob.(FGSM)|Clean|Avg Rob.(PGD, CW, SPSA, LGV, AutoAttack)|Search Cost (GPU sec)|
> > > |Clean one-shot|DrNAS|94.64|13.96|86.45|55.60|46857|
> > > |Robust one-shot|AdvRush|94.80|16.17|85.98|55.57|251245|
> > > |Clean zero-shot|GradNorm|92.84|15.55|81.61|51.61|9740|
> > > |Clean zero-shot|SynFlow|90.41|10.59|77.08|52.96|10138|
> > > |Robust zero-shot|CRoZe|94.45|**22.38**|85.05|**55.77**|17066|
> > >
> > > [1] Zero-Cost Proxies for Lightweight NAS\
> > > [2] Neural Architecture Design and Robustness: A Dataset\
> > > [3] NAS-Bench-201: Extending the Scope of Reproducible Neural Architecture Search

---

> > > > ### Author Response · Authors · 2023-08-14
> > > >
> > > > **Q3. The details on the paper about this subset is unclear.**\
> > > > We randomly selected 500 architectures from NAS-Bench-201 to avoid obtaining biased subsets. (Line 257-259)
> > > >
> > > > Currently, there is no benchmark search space that is designed to evaluate the robustness of adversarially trained networks. Moreover, it is challenging to adversarially train all 15,625 candidate architectures of the NAS-Bench-201 search space, because adversarial training of a single architecture requires 8 GPU hours. Specifically, when we adversarially train all candidate architectures simultaneously with 8 GPUs in the NAS-Bench-201 search space, it would take 651 days. Therefore, we randomly selected a subset of the NAS-Bench-201 search space.
> > > >
> > > > We evaluate our proxy on the [1] benchmark search space (Table 1. Left), which contains the robust accuracies of each candidate architecture that is trained on a clean dataset. To further evaluate our proxy on architectures that are adversarially trained [2], we constructed a specialized subset to evaluate the robustness of adversarially trained architectures and presented the results in Table 1. Right.
> > > >
> > > > [1] Neural Architecture Design and Robustness: A Dataset\
> > > > [2] Towards Deep Learning Models Resistant to Adversarial Attacks
> > > >
> > > > ---
> > > >
> > > > **Q4. Are there any insights from the components selected by NAS?**\
> > > > We assume that you are wondering about the insights from the architectures selected by our proxy. **Architectures selected by our proxy tend to have small feature variance**, as shown in Section 4.4 (also summarized in the following table). Based on these analyses, we believe that **models capable of capturing perturbation-invariant features can achieve overall high robustness against diverse perturbations**. Furthermore, an architecture that is robust to certain types of perturbations, for example, PGD attacks, might be overfitted to that specific perturbation, resulting in relatively lower robustness against other types of perturbations, such as common corruptions (Line 318). Therefore, targeting diverse perturbations could ensure robustness against currently unobserved types of perturbations.
> > > >
> > > > If this response does not address your question, could you please specify which 'components' you are referring to in your question? We would be delighted to provide further clarification.
> > > >
> > > > |Method|Feature variance| |
> > > > |-|-|-|
> > > > | |Standard-trained|Adversarially-trained|
> > > > |Architecture with the highest CRoZe score|**0.1489**|**0.1389**|
> > > > |Architecture with the best FGSM performance|0.1502|0.2100|
> > > > |Architecture with the best PGD performance|0.1699|0.3447|

---

> > > > > ### Author Response · Authors · 2023-08-16
> > > > > **Gentle Remider**
> > > > >
> > > > > **Dear reviewer,**
> > > > >
> > > > > This is a gentle reminder for the discussion periods.
> > > > >
> > > > > We genuinely appreciate your feedback and concerns. We have made every effort to address your questions in our previous responses. If you find our explanations satisfactory, we kindly ask you to reflect them in the score. Thanks again for the time and effort in reviewing our paper.
> > > > >
> > > > > Best,\
> > > > > Author

---

> > > > > > ### Author Response · Authors · 2023-08-18
> > > > > > **Reminder: Please check our previous responses**
> > > > > >
> > > > > > Dear Reviewer,
> > > > > >
> > > > > > Given that we have a limited window of **3 days remaining for further discussions**, we kindly request you to please review our responses along with the attached file. We have addressed all of your additional concerns and provide additional experimental results in our previous responses.  We sincerely hope to discuss our work and look forward to your constructive comments.
> > > > > >
> > > > > > For your convenience, we provide the **short summary** of our previous responses to your additional concerns. Detailed responses can be found in the previous responses.
> > > > > >
> > > > > > Best,\
> > > > > > Author
> > > > > >
> > > > > > ---
> > > > > >
> > > > > > > ### Summary
> > > > > > >- **Are there any citations on safety-critical applications for NAS?**
> > > > > > >   - We provided detailed references in our previous response. Reference [13] highlights the security risks associated with NAS, while [15] demonstrates the vulnerability of NAS to various perturbations. References [10-12] underscore the need for robust NAS.
> > > > > > >- **Could the authors elaborate on why the main tables are focusing on correlation?**
> > > > > > >   - We can replace the main table with the accuracy performance from Table 3 and R5 in the PDF file. However, it’s important to note that correlation is a critical metric, extensively used in train-free NAS, to evaluate the effectiveness of proxy design across a vast architecture search space.
> > > > > > >- **The details on the paper about this subset is unclear.**
> > > > > > >   - A subset of 500 is randomly selected from NAS-Bench-201 (Lines 257-259) to reduce computational costs of constructing robust search space when evaluating our proxy in an adversarially trained setting.
> > > > > > >- **Are there any insights from the components selected by NAS?**
> > > > > > >   - The architectures chosen by our proxy tend to **exhibit low feature variance**, as detailed in Section 4.4 (also summarized in the subsequent table). Based on this analysis, we posit that models which capture perturbation-invariant features are likely to demonstrate heightened robustness against a variety of perturbations.
> > > > > > |Method|Feature variance| |
> > > > > > |-|-|-|
> > > > > > | |Standard-trained|Adversarially-trained|
> > > > > > |Architecture with the highest CRoZe score|**0.1489**|**0.1389**|
> > > > > > |Architecture with the best FGSM performance|0.1502|0.2100|
> > > > > > |Architecture with the best PGD performance|0.1699|0.3447|

---

> > > > > > > ### Comment · Reviewer_KEHv · 2023-08-19
> > > > > > > **Response**
> > > > > > >
> > > > > > > Dear authors,
> > > > > > >
> > > > > > > I am thankful for your response. The answers Q3 and Q4 are clear to me, thank you for providing them. I am looking forward to the discussion with the other reviewers.
> > > > > > >
> > > > > > >
> > > > > > > Are all the percentages reported (and especially the new experiments added in the rebuttal) single runs?

---

> > > > > > > > ### Author Response · Authors · 2023-08-19
> > > > > > > > **Responses are Uploaded**
> > > > > > > >
> > > > > > > > Dear Reviewer,
> > > > > > > >
> > > > > > > > We are sincerely glad that your concerns about Q3 and Q4 are resolved.
> > > > > > > >
> > > > > > > > For your additional question, **all the experimental results conducted within the NAS-Bench-201 search space are unaffected by the number of trials**, encompassing Spearman’s rank correlation (Tables 1, 2, 4, R1, R3, and R6) and final clean/robust accuracy (Table R5).
> > > > > > > >
> > > > > > > > Our proxy consistently identifies the same architecture over multiple runs on the NAS-Bench-201 search space. That is, the candidate architecture with the highest proxy score does not change over runs. This is further reinforced by previous studies [1] and [2], where the performance of NASWOT on the NAS-Bench-201 search space (Table 2 in [1]) and all of the experimental results presented in [2] are reported without variance.
> > > > > > > >
> > > > > > > > However this is not the case for DARTS search space, since we are unable to fully acquire scores for the large number of $10^{19}$ potential candidate architectures, and thus we use sampling to search for the candidate. Consequently, adhering to your insights, **we provide the mean and standard variation for Table 3 based on three independent search trials with SynFlow, GradNorm, and CRoZe on DARTS search space** in the following table. As shown in the Table, **CRoZe consistently outperforms with a small variance**.
> > > > > > > >
> > > > > > > > |NAS Type|Proxy|Clean|CC.|FGSM|
> > > > > > > > |-|-|-|-|-|
> > > > > > > > |Clean one-shot|DrNAS| 94.64 | 72.62 | 13.96 |86.45|24.33|
> > > > > > > > |Robust one-shot|AdvRush| 94.80 | 72.00 | 16.17 |27.63|
> > > > > > > > |Clean zero-shot|GradNorm|92.95 ($\pm$0.67) | 63.03 ($\pm$6.33) | 15.64 ($\pm$0.15)|
> > > > > > > > |Clean zero-shot|SynFlow|92.15 ($\pm$1.25) | 70.74 ($\pm$2.69) | 12.65 ($\pm$1.46)|
> > > > > > > > |Robust zero-shot|CRoZe|94.34 ($\pm$0.08) | 74.11 ($\pm$1.40) | 20.51 ($\pm$1.32) |
> > > > > > > >
> > > > > > > > We are committed to delivering updated results that encompass the mean and standard deviation from three separate trials for Table R2. However, it will take more time to conduct multiple runs. We deeply appreciate your feedback which helps strengthen the comprehensiveness and reliability of our work.
> > > > > > > >
> > > > > > > > Based on your feedback, it appears that you still have concerns over Q1 and Q2. Could you specify the issues related to these points? If Q1 and Q2 underpin the score of 4, we are eager to address them comprehensively during the discussion phase.
> > > > > > > >
> > > > > > > > Best,\
> > > > > > > > Author
> > > > > > > >
> > > > > > > > [1] Generalization Properties of NAS under Activation and Skip Connection Search\
> > > > > > > > [2] Zero-Cost Proxies for Lightweight NAS

---

> > > > > > > > > ### Author Response · Authors · 2023-08-21
> > > > > > > > > **Final Reminder**
> > > > > > > > >
> > > > > > > > > Dear reviewer,
> > > > > > > > >
> > > > > > > > > **With only a few hours left in the discussion period, we kindly remind you to review our response.**\
> > > > > > > > > We respectfully request that you check our response within this timeframe.\
> > > > > > > > > If there are any remaining concerns, please let us know.
> > > > > > > > >
> > > > > > > > > Best,\
> > > > > > > > > Author

---

> > > > > > > > > > ### Comment · Reviewer_KEHv · 2023-08-21
> > > > > > > > > > **Response**
> > > > > > > > > >
> > > > > > > > > > Dear authors,
> > > > > > > > > >
> > > > > > > > > > I am not sure how the correlations are unaffected by the number of trials. If the ranking of the architectures changes between runs (which happens often in my experience with NAS in larger search spaces), how is the correlation not affected?
> > > > > > > > > >
> > > > > > > > > > Also, personally I do not find it convincing to have a paper on NAS and then sample only a handful of architectures from larger search spaces. This beats the purpose of NAS, and on top it does not perform favorably to standard networks.

---

> > > > > > > > > > > ### Author Response · Authors · 2023-08-21
> > > > > > > > > > > **Dear reviewer,**
> > > > > > > > > > >
> > > > > > > > > > > Dear reviewer,
> > > > > > > > > > >
> > > > > > > > > > > We kindly hope the following responses may resolve your concerns. If we did not correctly interpret or address them, please let us know.
> > > > > > > > > > >
> > > > > > > > > > > > I am not sure how the correlations are unaffected by the number of trials. If the ranking of the architectures changes between runs (which happens often in my experience with NAS in larger search spaces), how is the correlation not affected?
> > > > > > > > > > >
> > > > > > > > > > >
> > > > > > > > > > > * We apologize for the confusion in our previous response. The final correlation performance **remains unchanged when rounded to the third decimal point due to the extremely small variance (4.33e-05)** in the relatively small NAS-Bench-201 search space. We recognize your observation that in larger search spaces, as you've experienced with NAS, correlations might display variances due to densely clustered architectural candidates. However, please note that, currently, the NAS-Bench-201 search space is **the only one with ground truth robust accuracies for all models** to calculate the correlations. As a result, we cannot evaluate the variance of correlation in larger spaces at this time.
> > > > > > > > > > > * Regarding the variance in final performance, as previously noted, since we select the same architecture as top-1 within three trials in NAS-Bench-201, the final performance remains consistent.
> > > > > > > > > > >
> > > > > > > > > > > ----
> > > > > > > > > > > > Also, personally I do not find it convincing to have a paper on NAS and then sample only a handful of architectures from larger search spaces. This beats the purpose of NAS, and on top it does not perform favorably to standard networks.
> > > > > > > > > > >
> > > > > > > > > > >
> > > > > > > > > > > * To clarify, are you expressing concerns about the efficacy of zero-cost NAS compared to the one-shot based NAS approach?
> > > > > > > > > > > * Zero-cost NAS, as detailed in previous works [1,2,3], offers the advantage of significantly reduced computational cost while still being able to identify architectures that are comparable to those discovered using the one-shot based NAS approach [4,5]. In line with similar research, when the objective is to identify architectures robust against a variety of perturbations, the search cost for one-shot based NAS increases considerably—directly proportional to the number and complexity of the perturbation types. In contrast, **our proxy allows for the discovery of robust architectures with reduced search costs, achieving either better or at least comparable robustness** (Figure 2).
> > > > > > > > > > > * We strongly believe that exceeding computational costs associated with NAS are a significant bottleneck. From this perspective, our approach introduces a valuable contribution, enabling a more efficient search for robust architectures.
> > > > > > > > > > >
> > > > > > > > > > > ----
> > > > > > > > > > > If we did not correctly understand your concerns, please let us know. We appreciate the feedback and are open to further discussions to enhance the clarity and relevance of our work.
> > > > > > > > > > >
> > > > > > > > > > > ----
> > > > > > > > > > > [1] Zero-Cost Proxies for Lightweight NAS, ICLR 2021 \
> > > > > > > > > > > [2] Neural Architecture Search without Training, ICML 2021 \
> > > > > > > > > > > [3] Nasi: Label-and data-agnostic neural architecture search at initialization, ICLR 2022 \
> > > > > > > > > > > [4] PC-DARTS: Partial Channel Connections for Memory-Efficient Architecture Search, ICLR 2020 \
> > > > > > > > > > > [5] Once-for-All: Train One Network and Specialize it for Efficient Deployment, ICLR 2020

---

### Official Review · Reviewer_41nW · 2023-07-05

**Soundness:** 3 good
**Presentation:** 3 good
**Contribution:** 3 good
**Rating:** 6
**Confidence:** 4

**Summary:**

The paper proposes a lightweight approach to generate new architectures with robustness formulated in the NAS process.
The paper claims that the proposed method is capable of generating architectures that can learn generalized features with higher robustness.

**Strengths:**

Efficient algorithms for generating robust architectures are an important topic in NAS field and the paper focuses on an important topic.
The proposed method is simple but effective based on the experimental results.

**Weaknesses:**

While I like the approach I think the experimental results are limited to show the effectiveness of the proposed method properly.
1- Most of the experiments are based on the FGSM and as mentioned in the paper it is considered as the worst adversarial attack currently.
2- I can see one experiment on PGD in the paper but it is very limited.
3- I was hoping to see some comparison with the well-known human-made architectures (like ResNet) for both clean and robust accuracy.
4-  Having more comprehensive experiments on more SOTA adversarial attacks like AutoAttack would help to show the effectiveness of the proposed method better.
5- It has been claimed that the proposed method is "Generalized to Diverse Perturbations" but I think there is not enough evidence and more experiments should have been done to make it justice.

**Questions:**

The main question is whether it is possible to provide a more comprehensive evaluation to show the effectiveness of the proposed method.

**Limitations:**

Based on the current experimental results it seems the effectiveness of the proposed method is very limited

---

> ### Author Rebuttal · Authors · 2023-08-06
>
> **W1, 2. Experimental results are limited to FGSM and PGD.**
> - First, we would like to clarify that we report experimental results on three types of perturbations, including **FGSM, PGD, and 16 types of common corruption (Table 1, 2, 3)**. However, following your suggestion, we further provide additional experimental results against a wide range of recent adversarial attacks, including **CW, DeepFool, SPSA, LGV, and AutoAttack** on the NAS-Bench-201 search space in **Table R1 in the PDF file**. Specifically, for the accurate evaluation of the robust accuracy, we adversarially train each candidate architecture in the NAS-Bench-201 search space on CIFAR-10 following the protocol from [3]. Then, we subsequently evaluate the robustness of those models against diverse adversarial attacks.
> - **Our proxy shows the highest overall correlation of 0.399 for robust accuracies** while the best baseline (GradNorm) achieves 0.352, which demonstrates that our proxy can rapidly search for a neural architecture capable of learning robust features against diverse attacks. (*Detailed table can be found in Table R1 in PDF*)
> |Proxy Type|FGSM|PGD|CW|DeepFool|SPSA|LGV|AutoAttack|Avg.|
> |-|-|-|-|-|-|-|-|-|
> |FLOPs|0.357|0.446|0.189|0.364|0.196|0.347|0.365|0.323|
> |GradNorm|0.378|0.446|**0.264**|0.421|0.149|0.401|0.405|0.352|
> |NASWOT|0.311|0.354|0.240|0.250|0.197|0.265|0.280|0.271|
> |CRoZe|**0.441**|**0.532**|0.220|**0.454**|**0.240**|**0.449**|**0.458**|**0.399**|
>
> ------------
> **W3. Comparison with human-made architecture is needed.**
> - Following your valuable suggestion, we additionally provide comparisons with well-known human-made architectures with **similar parameter sizes, such as Conv4, Conv6, MobileNetV2, and ResNet12**. The neural architectures identified by our proxy even show **better clean and robust accuracy with fewer parameters than ResNet12**.
> - Specifically, our method achieves **2.11% and 2.35% higher clean and robust accuracies (PGD-20) with 2.48MB fewer parameters compared to ResNet12**. All models are adversarially trained following the conventional protocol [1] on CIFAR-10 and evaluated against PGD [1]. (*Detailed table can be found in Table R4 in PDF*)
> | | # Params (MB)|Clean|PGD-20|HRS|
> |-|-|-|-|-|
> |Conv4|0.03|60.12|29.98|40.01|
> |Conv6|0.05|65.92|33.04|44.02|
> |MobileNetV2|2.24|66.04|33.04|46.79|
> |ResNet12|8.00|82.94|49.69|62.15|
> |CRoZe|5.52|**85.05**|**52.04**|**64.57**|
>
> ----------
> **W4. Experimental results against AutoAttack would be helpful to demonstrate its effectiveness.**
> - We evaluated against SOTA adversarial attacks, including **AutoAttack, SPSA, LGV, DeepFool, and CW in Table R2 and Figure R1 in the PDF file**.  All architecture searched by each method on DARTS search space is adversarially trained following [1] on CIFAR-10.
> - Architecture that is identified by our proxy shows **the highest average robust accuracy of 55.77%** against various adversarial perturbations, even with **15 times more efficient search cost** compared to the AdvRush.
> |NAS Type|Method|Clean|PGD-20|CW|SPSA|LGV|AutoAttack|Avg. Rob.|Search Cost (GPU sec)|
> |-|-|-|-|-|-|-|-|-|-|
> |Clean one-shot|DrNAS|86.45|54.66|11.39|85.73|79.82|52.40|55.60|46857|
> |Robust one-shot|AdvRush|85.98|53.89|6.68|80.85|79.61|51.88|55.57|251245|
> |Clean zero-shot|GradNorm|81.61|49.86|12.02|77.19|73.27|46.69|51.61|9740|
> |Clean zero-shot|SynFlow|77.08|45.95|26.50|75.78|74.14|42.45|52.96|10138|
> |Robust zero-shot|CRoZe|85.05|52.04|16.82|83.23|77.62|49.15|**55.77**|17066|
>
> -----------------------------------
> **W5. Not enough evidence for claim of "Generalized to Diverse Perturbations".**
> - We clarify that our model demonstrates its ability to generalize to **7 types of adversarial attacks (Table R1, R2 in the PDF file) and 16 types of common corruption perturbations (Table 1, 2) with diverse search space**. Especially, simple version of the results of Spearman's rank correlation evaluation with adversarially-trained NAS-Bench-201search space against recent adversarial attacks (Table R1) is presented in the below.
> - Therefore, we believe our approach is effective in searching for neural architecture that is capable of learning generalizable robust features against diverse perturbations.
> - Through extensive experiments on NAS-Bench-201 and DARTS search space against a wide range of perturbations in both standard (Table 1,2) and adversarial (Table 1, R1, R2) training settings, we validate our approach, as noted by reviewer m6P2.
> |Proxy Type|CW|DeepFool|SPSA|LGV|AutoAttack|Avg.|
> |-|-|-|-|-|-|-|
> |GradNorm|**0.264**|0.421|0.149|0.401|0.405|0.352|
> |NASWOT|0.240|0.250|0.197|0.265|0.283|0.271|
> |CRoZe|0.220|**0.454**|**0.240**|**0.449**|**0.458**|**0.399** |
>
> -----------------------------------
> **Question/Limitation. A more comprehensive evaluation is necessary to validate the effectiveness of the approach.**
> - Thanks to your suggestions, in addition to FGSM, PGD, and common corruption (Table 1,2,3), we further conducted experiments on a wide range of attacks including **CW, DeepFool, SPSA, LGV, and AutoAttack** on NAS-Bench-201 search space (Table R1) and DARTS search space (Table R2).
> - Moreover, we demonstrate that our proxy can rapidly search for neural architectures capable of learning generalizable representations against diverse perturbations in both standard (Table 1 left, 3,2) and adversarial settings (Table 1 right, R1, R2).
>
> We strongly believe that these additional experimental results show the clear effectiveness of our method against a diverse set of attacks, which further strengthens our work and its potential practical impact.
>
> Thank you for the valuable suggestion.
>
> [1] Madry et al., Towards Deep Learning Models Resistant to Adversarial Attacks, ICLR 2018 \
> [2] Croce et al., Reliable Evaluation of Adversarial Robustness with an Ensemble of Diverse Parameter-free Attacks, ICML 2020\
> [3] Wong et al., Fast is better than free: Revisiting adversarial training, ICLR 2020

---

> > ### Author Response · Authors · 2023-08-17
> > **Gentle Reminder**
> >
> > Dear Reviewer,
> >
> > This is a gentle reminder for the discussion period.
> >
> > We have addressed all of your initial concerns and provide additional experimental results in our previous responses.
> >
> > We sincerely hope to discuss our work and look forward to your constructive comments. For your convenience, we provide the **short summary** of our previous responses to your initial concerns. Detailed responses can be found in the previous responses.
> >
> > Best,\
> > Author
> >
> > ---
> >
> > > ### Summary
> > >- **Experimental results are limited to FGSM and PGD. Experimental results against AutoAttack are needed.**
> > >   - (Shown through additional experiment in Table R1) Our method achieves **the highest overall correlation of 0.399 for average robust accuracies (CW, DeepFool, SPSA, LGV, and AutoAttack)** while the best baseline shows 0.352.
> > >   - (Shown through additional experiment in Table R2, Figure R1) Our proxy demonstrates **the highest average robustness of 55.77%** against various attacks (CW, LGV, SPSA, and AutoAttack) with **15 times less search cost** compared to baseline.
> > >- **Comparisons with human-made architecture are needed.**
> > >   - (Shown through additional experiment in Table R4) Under adversarial training, our method achieves **2.11% and 2.35% higher clean and robust accuracies** (PGD-20) with 2.48MB fewer parameters compared to ResNet12.
> > >- **Not enough evidence for the claim of ‘generalized to diverse perturbations’.**
> > >   - (Shown through additional experiment in Table R1, R2, Figure R1, Table 1, 2, 3) We clarify that our model demonstrates its ability to generalize to **7 types of adversarial attacks** including FGSM, PGD, CW, DeepFool, SPSA, LGV, and AutoAttack (Table R1, R2, Figure R1) and **16 types of common corruption perturbations** (Table 1, 2, 3) with diverse search space (NAS-Bench-201, DARTS) under both standard (Table 1 Left, 2) and adversarial training (Table 1 Right, R1, R2) settings.

---

> > > ### Author Response · Authors · 2023-08-18
> > > **Reminder: Please check our previous responses**
> > >
> > > Dear Reviewer,
> > >
> > > We sincerely value the time and effort you have dedicated to reviewing our paper.
> > >
> > > We understand and appreciate the volunteer effort and time you’ve given to ensure a fair and constructive review process for NeurIPS. Given that **we have only 3 days remaining for further discussions**, we kindly request that you review our responses and the attached file. **To save you time, we have also provided a summarized response in our previous communication.**
> > >
> > > We are confident that we have addressed all of your concerns by providing additional experimental results and explanations. Therefore, we kindly ask you to incorporate these updates into your final review and score. Your consideration in this matter is highly appreciated.
> > >
> > > We eagerly look forward to your insightful comments and feedback.
> > >
> > > Best regards,\
> > > Author

---

> > > > ### Author Response · Authors · 2023-08-20
> > > > **Gentle Reminder**
> > > >
> > > > Dear Reviewer,
> > > >
> > > > **With just 1 day left in our discussion period**, we politely ask you to review our recent responses and the attached file showcasing the additional experimental results you requested. For your convenience, **a summary also has been provided in our previous response.**
> > > >
> > > > We truly appreciate your commitment to the review process and await your valuable feedback.
> > > >
> > > > Warm regards,\
> > > > Author

---

> > > > > ### Author Response · Authors · 2023-08-21
> > > > > **Final Reminder**
> > > > >
> > > > > Dear reviewer,
> > > > >
> > > > > **With only a few hours left in the discussion period, we kindly remind you to review our response.**\
> > > > > We respectfully request that you check our response within this timeframe.\
> > > > > If there are any remaining concerns, please let us know.
> > > > >
> > > > > Best,
> > > > > Author

---

### Official Review · Reviewer_DQhw · 2023-07-06

**Soundness:** 3 good
**Presentation:** 3 good
**Contribution:** 3 good
**Rating:** 6
**Confidence:** 4

**Summary:**

This work proposes a new zero-shot proxy to find robust NN architecture at initialization. The proxy utilizes the consistency of model features and gradients for clean and perturbed input. Experiments are conducted on robust NAS benchmarks. Performance is also provided for end-to-end NAS on DARTS search space.

**Strengths:**

1. This work provides a novel and effective proxy for the search of robust NN model
2. The proposed method is well formulated and easy to follow
3. Results are provided on both NASBench and end-to-end search to show the effectiveness of the proposed method

**Weaknesses:**

1. The theortical insight behind the proposed consistency score is unclear. As the score is only computed at the initialization of the model, it is unclear whether the proposed score is consistent for different random weight initialization, and if its correlation to the model robustness is affected by different adversarial training methods. More theortical analysis or ablation studies are needed to answer these questions
2. End-to-end NAS performance is an important metric to show the true effectiveness of the proposed method. However, in Tab.3 the results are limited to clean training and robustness against FGSM perturbation only. More final performance results with adversarial training and different types of perturbation is needed to verify the contribution.

**Questions:**

See weakness.

**Limitations:**

The work is limited in theoretical insights of the proposed method. No potential negative socal impact is observed.

---

> ### Author Rebuttal · Authors · 2023-08-06
>
> **W1. The theoretical insight is unclear and the ablation experiments on different weight initialization methods and adversarial training methods are needed.**
>
> - The underlying theoretical insight of our proxy is premised on the notion that **a robust model should learn invariant useful features with respect to the clean and perturbed images [1,2,8,9]**. Based on this theoretical insight, we propose to design the proxy to measure the similarity of features, gradients, and parameters between the clean and perturbed inputs.
> - In response to your concerns regarding the compatibility of our proxy with various weight initialization methods, we conducted additional experiments with **Random initialization, Kaiming initialization [5], and Xavier initialization [6]** on the NAS-Bench-201 search space on CIFAR-10 in **Table R3 in the PDF file**.  As demonstrated, **our proxy maintains a consistently higher correlation** compared to the baselines, **irrespective of the weight initialization methods employed**. Specifically, our proxy shows the highest average correlation of **0.568 and 0.399 for standard training and adversarial training scenario, respectively, while the best baseline only achieves 0.529 and 0.352** against diverse perturbations with the same random weight initialization. Since our approach considers the similarity vector of the parameters and gradients between the clean and perturbed image, Our superior performance can be achieved regardless of the initialization method. (*Detailed table with more baselines can be found in Table R1 and R3 in the PDF*)
> | |Standard-Trained| | | | |Adversarially-Trained| | | | | | | | |
> |-|-|-|-|-|-|-|-|-|-|-|-|-|-|-|
> | |Clean|FGSM|PGD|CC|Avg.|FGSM|PGD|CW|DeepFool|SPSA|LGV|AutoAttack|Avg.|
> |FLOPs|0.726|0.753|0.183|0.384|0.512|0.357|0.446|0.189|0.364|0.196|0.347|0.365|0.323|
> |SynFlow|0.777|0.778|0.163|0.396|0.529|0.369|0.442|0.202|0.397|0.196|0.387|0.383|0.339|
> |GradNorm|0.638|0.750|0.259|0.383|0.508|0.378|0.446|0.264|0.421|0.149|0.401|0.405|0.352|
> |NASWOT|0.660|0.511|-0.280|0.206|0.274|0.311|0.354|0.240|0.250|0.197|0.265|0.280|0.271|
> |CRoZe(Random)|0.823|0.826|0.188|0.436|**0.568**|0.441|0.532|0.220|0.454|0.240|0.449|0.458|**0.399**|
> |CRoZe(Kaiming)|0.812|0.818|0.189|0.430|0.562|0.428|0.512|0.217|0.443|0.227|0.426|0.436|0.384|
> |CRoZe(Xavier)|0.816|0.822|0.190|0.433|0.565|0.428|0.513|0.217|0.442|0.227|0.425|0.436|0.384|
>
> - With regard to adversarial training, unfortunately, **no benchmark datasets currently exist that contain robust accuracy of models that are trained with diverse adversarial training methodologies**, such as AT [3] and TRADES [4]. Therefore, it is challenging to illustrate dependencies on the type of adversarial training methods. However, based on the RobustBench [7], we can presume **different adversarial training methods may not affect the correlations** because the rank of performance holds even in diverse types of architectures (WideResNet 70-16, WideResNet 34-10, and ResNet18). Additionally, the performance ranking between our method and zero-cost proxy baselines remains consistent across different adversarial training types [3, 4], as shown in the following table.
> |Proxy Type|Adversarial Training Type|Clean|PGD-20|Rank|
> |-|:-:|:-:|:-:|:-:|
> |GradNorm|[3]|81.61|49.86|2|
> | |[4]|72.43|44.84|2|
> |SynFlow|[3]|77.08|45.95|3|
> | |[4]|59.87|36.07|3|
> |CRoZe|[3]|85.05|52.04|1|
> | |[4]|79.48|51.89|1|
>
> [1] Xia et al., Feature Denoising for Improving Adversarial Robustness, CVPR 2019 \
> [2] Ilyas et al., Adversarial Examples Are Not Bugs, They Are Features, NeurIPS 2019 \
> [3] Madry et al., Towards Deep Learning Models Resistant to Adversarial Attacks, ICLR 2018 \
> [4] Zhang et al., Theoretically Principled Trade-off between Robustness and Accuracy, ICML 2019 \
> [5] He et al., Delving Deep into Rectifiers: Surpassing Human-Level Performance on ImageNet Classification, IEEE 2015 \
> [6] Glorot et al., Understanding the difficulty of training deep feedforward neural networks, JMLR 2010 \
> [7] Croce et al.,  RobustBench: a standardized adversarial robustness benchmark, NeurIPS 2021 \
> [8] Goodfellow et al., Explaining and Harnessing Adversarial Examples, ICLR 2015 \
> [9] Zhang et al., Understanding deep learning requires rethinking generalization, ICLR 2017
>
> ----------
>
> **W2. End-to-end performance is limited to clean training and robustness against FGSM attack.**
>
> - We appreciate your thoughtful response. In light of your comments, we have included additional experimental results that demonstrate the end-to-end NAS performance within the DARTS search space, where the searched architectures are **adversarially trained on CIFAR-10** following [1] and are subsequently evaluated against diverse perturbations including **PGD, CW, SPSA, LGV, and AutoAttack in Table R2 in the PDF file**. Our proxy, as substantiated by the results, shows **superior robustness, averaging robustness of 55.77%** against diverse perturbations, even with **15 times more efficient search cost** compared to the existing Robust NAS method (AdvRush). (*Detailed table can be found in Table R2 and Figure R1 in the PDF*)
> |NAS Type|Method|Clean|PGD-20|CW|SPSA|LGV|AutoAttack|Rob. Avg.|Search Cost (GPU sec)|
> |-|-|-|-|-|-|-|-|-|-|
> |Clean one-shot|DrNAS|86.45|54.66|11.39|85.73|79.82|52.40|55.60|46857|
> |Robust one-shot|AdvRush|85.98|53.89|6.68|80.85|79.61|51.88|55.57|251245|
> |Clean zero-shot|GradNorm|81.61|49.86|12.02|77.19|73.27|46.69|51.61|9740|
> |Clean zero-shot|SynFlow|77.08|45.95|26.50|75.78|74.14|42.45|52.96|10138|
> |Robust zero-shot|CRoZe|85.05|52.04|16.82|83.23|77.62|49.15|**55.77**|17066|
>
> [1] Madry et al., Towards Deep Learning Models Resistant to Adversarial Attacks, ICLR 2018
>
> ---------------------------------------------------
> **Limitation. No potential negative social impact is observed.**
> - Since our approach does not bring any potential negative social impact, we neglect it as mentioned in the Neurips 2023 Author policy.

---

> > ### Author Response · Authors · 2023-08-17
> > **Gentle Reminder**
> >
> > Dear Reviewer,
> >
> > This is a gentle reminder for the discussion period.
> >
> > We have addressed all of your initial concerns and provide additional experimental results in our previous responses.
> >
> > We sincerely hope to discuss our work and look forward to your constructive comments. For your convenience, we provide the **short summary** of our previous responses to your initial concerns. Detailed responses can be found in the previous responses.
> >
> > Best,\
> > Author
> >
> > ---
> > > ### Summary
> > >- **The theoretical insight is unclear.**
> > >   - The underlying theoretical insight of our proxy is premised on the notion that a robust model should learn invariant useful features with respect to the clean and perturbed images, which is motivated by previous works. (Please visit our initial response with a detailed explanation.)
> > >- **Ablation study on weight initialization methods and adversarial training methods are needed.**
> > >   - (Shown through additional experiment in Table R3) Our proxy maintains a consistently higher correlation compared to the baselines regardless of the weight initialization methods.
> > >   - (In previous response) Based on the RobustBench [7] and our experimental results, we can presume different adversarial training methods may not affect the correlations.
> > >- **End-to-end performance is limited to clean training and robustness against FGSM attack**
> > >   - (Shown through additional experiment in Table R2, Figure R1) Under adversarial training scenario, we demonstrate that our proxy can identify robust architectures with the **highest average robust accuracy of 55.77% against PGD, CW, SPSA, LGV, and AutoAttack with 15 times less search cost** compared to baseline.

---

> ### Comment · Reviewer_DQhw · 2023-08-17
> **Thanks for the response**
>
> I would like to thank the author for the detailed responses. The response resolves my concern with the stability and effectiveness of the proposed method. I will increase my rating to the paper to weak accept.
>
> One remaining question about the provided results on training with different adversarial training methods. As I understand the consistency between the performance ranking, I'm not sure why TRADES appears to be worse in both clean and robust accuracy than PGD for all the cases. Is this due to some hyperparameter choices?

---

> > ### Author Response · Authors · 2023-08-17
> >
> > **Dear Reviewer,**
> >
> > We sincerely appreciate the time and effort you dedicated to reviewing our work. We are pleased to note that our responses have addressed your initial concerns.
> >
> > In light of your feedback, we adversarially trained the cell-based candidate architectures using the same hyper-parameter settings as stacked-based architectures, as referenced in [1]. However, it's worth noting that determining the optimal hyper-parameters for the adversarial training of cell-based architectures using the TRADES method [2] might differ from those of stacked architectures. This discrepancy can lead to reduced performance in both clean accuracy and robustness. It can be challenging to find the optimal hyper-parameters for cell-based architectures. Despite these challenges, we attempted hyper-parameter tuning in a manner similar to [1] for cell-based architectures using TRADES. We will keep you updated on our progress and share our results as soon as possible.
> >
> > Thank you once again for your invaluable feedback.
> >
> > Best regards,\
> > Author
> >
> > [1] Pang et al., Bag of Tricks for Adversarial Training, ICLR 2021\
> > [2] Zhang et al., Theoretically Principled Trade-off between Robustness and Accuracy, ICML 2019

---

### Official Review · Reviewer_m6P2 · 2023-07-10

**Soundness:** 3 good
**Presentation:** 3 good
**Contribution:** 3 good
**Rating:** 7
**Confidence:** 4

**Summary:**

This work introduces a lightweight proxy, CRoZe, designed to facilitate the development of Neural Architecture Search (NAS) based architectures that are robust across a diverse set of semantic-preserving perturbations. CRoZe operates by measuring consistency across the features, parameters, and gradients for a given clean image and its perturbed counterpart. This approach enables it to withstand a wide range of perturbations, unlike existing methods that focus on a specific set of perturbations (adversarial or Out-of-Distribution samples). Experimental results demonstrate that the proposed proxy can rapidly and efficiently search for neural architectures that exhibit consistent robustness against various perturbations across multiple benchmark datasets (CIFAR-10, CIFAR-100, ImageNet16-120) and diverse search spaces (NAS Bench201, DARTS). This performance significantly surpasses that of existing clean zero-shot NAS and robust NAS methods, all while reducing search costs.

**Strengths:**

* The focus on developing architectures that are robust against a *diverse range of perturbations* is novel and has not been studied earlier, presenting a wide range of potential applications.
* The problem setting is well-motivated and effectively approached, with the introduction setting the context particularly well.
* The experimental setup is solid, employing two benchmarks and three datasets.

**Weaknesses:**

* The choice of adversarial attacks -- FGSM and PGD -- seems somewhat outdated from the perspective of adversarial attacks. It would be beneficial to understand how the proposed method would perform with more recent adversarial attack methods such as  [LGV](https://arxiv.org/abs/2207.13129), [SPSA](https://arxiv.org/abs/1802.05666) etc.
* The same concern applies to the set of experiments where adversarial training is used. The choice of $L_\infty$ PGD attacks may not be very contemporary, and exploring more recent approaches could enhance the paper's relevance and applicability.

**Questions:**

1. How does the proposed method perform when confronted with novel types of perturbations that were not considered during the architecture search? Is there a mechanism to make the method more adaptive to new perturbations? (This question is related to Weakness 1)
2.  Is there a specific reason for choosing a relatively older search space like DARTS, especially when there are more recent search spaces available? (Please refer to NAS Bench  [NAS Bench](https://www.automl.org/nas-overview/nasbench/), [$A^{3}D$: A Platform of Searching for Robust Neural Architectures and  Efficient Adversarial Attacks](https://arxiv.org/pdf/2203.03128))
3. Comparison with robust ensemble-based methods: Ensemble-based methods in NAS are often known to be more robust. Does the proposed proxy support ensemble-based approaches?

**Limitations:**

Limitation and Broader impact are discussed in the supplementary

---

> ### Author Rebuttal · Authors · 2023-08-07
>
> **W1. Evaluation against recent adversarial attacks is needed (i.e., LGV, SPSA).**
> * Thanks for your comment, we provide additional experimental results on recent adversarial attacks such as CW, DeepFool, SPSA, LGV, and AutoAttack by evaluating Spearman’s rank correlation on the NAS-Bench-201 search space where each candidate architecture is adversarially trained with PGD-7 on CIFAR-10.
> * For evaluation, we employ CW and DeepFool attack with 50 steps, SPSA with  $\epsilon$=8/255 and a single iteration, LGV with $\epsilon$=4/255, $\alpha$=4/255/10, and 10 steps, and AutoAttack with $\epsilon$=8/255.
> * As evidenced by the correlation in the ‘Avg.’ column, our proxy consistently **shows the highest overall correlation of 0.399 for robust accuracies** even against recent adversarial attacks, while the best baseline (GradNorm) only achieves 0.352. This suggests that our proxy is capable of identifying neural architectures that are robust against a wide range of adversarial attacks. (Detailed table can be found in Table R1 in PDF)
> |Proxy Type|CW|DeepFool|SPSA|LGV|AutoAttack|Avg.|
> |-|-|-|-|-|-|-|
> |GradNorm|**0.264**|0.421|0.149|0.401|0.405|0.352|
> |NASWOT|0.240|0.250|0.197|0.265|0.283|0.271|
> |CRoZe|0.220|**0.454**|**0.240**|**0.449**|**0.458**|**0.399** |
>
> ------------------
>
> **W2. The choice of PGD attacks may not be very contemporary.**
> * The main reason for using adversarial training is that it is the most straightforward way to attain a robust model for a reliable robust evaluation. To address your concern, we additionally provide the experimental results on the NAS-Bench-201 search space (our response to W1, **Table R1**) and the DARTS search space on CIFAR-10, presented in **Table R2** in the PDF file. We assess the robustness of the architectures identified by each method against recent adversarial attacks, including CW, DeepFool, LGV, SPSA, and AutoAttack. Our proxy underscores its ability to discover neural architectures demonstrating **the highest average robustness of 55.77% against various attacks with 15 times less search cost** compared to AdvRush. (Detailed table can be found in Table R2 in PDF)
> ||Type|Search Cost (GPU sec)|Avg. Rob.|
> |-|-|-|-|
> |DrNAS|Clean one-shot|46857|55.60|
> |AdvRush|Robust one-shot|251245|55.57|
> |GradNorm|Clean zero-shot|9740|51.61|
> |SynFlow|Clean zero-shot|10138|52.96|
> |CRoZe|Robust zero-shot|17066|**55.77**|
>
> -------------
>
> **Q1. How does the CRoZe perform against novel types of perturbations?**
> * Experimental results against novel attacks can be found in response to W1, W2.
> * Our theoretical insight is derived from that a robust model can learn invariant features with respect to clean and perturbed images regardless of the type of perturbations applied to input [1,2,3]. On top of this motivation, our proxy mainly focuses on the consistency of features, parameters, and gradients between clean and perturbed inputs without targeting a specific type of perturbation for assessing the robustness of the given neural architecture.
> * Due to properties not limited by perturbation types, our proxy works well with any kind of perturbed inputs such as Gaussian Noise or adversarial examples (Table 4). Furthermore, our proxy demonstrates the ability to handle novel types of perturbation not considered during the search (Table 1,2).
>
> [1]Xia et al., Feature Denoising for Improving Adversarial Robustness, CVPR 2019\
> [2]Ilyas et al., Adversarial Examples Are Not Bugs, They Are Features, NeurIPS 2019 \
> [3]Goodfellow et al., Explaining and Harnessing Adversarial Examples, ICLR 2015
>
> ------------
> **Q2. Is there a reason for choosing an older search space like DARTS?**
> * We choose the DARTS search space because 1) for a fair comparison with prior works [1,2] and 2) it is the largest search space for image classification tasks as shown in the following table. Besides, please note that we also conducted experiments on NASBench-201 in Table 1, and 2.
> |Benchmark|NAS-Bench-101|NAS-Bench-201|NAS-Bench-301|NAS-Bench-Macro|DARTS|
> |-|-|-|-|-|-|
> |Size|423k|6k | $10^{18}$ |6k| $10^{18}$ |
> * Furthermore, the search spaces (NASBench) that you have mentioned are, in fact, encapsulated within NAS-Bench-101, NAS-Bench-201, and NAS-Bench-301, differing only in their use of metrics for neural architectures outside of validation accuracy. In NAS-Bench-301, each neural architecture is defined with the configuration of the DARTS search space which eventually aligns with the DARTS search space. In addition, the $A^3$D search space is a subset of the DARTS search space with fewer operation choices, so compared to $A^3$D, we employ a wider range of operations, including sep_con_7x7 and dil_con_7x7.
>
> [1]Chen et al., DrNAS: Dirichlet Neural Architecture Search, ICLR 2021\
> [2]Mok et al., AdvRush: Searching for Adversarially Robust Neural Architectures, ICCV 2021
>
> ------------
> **Q3. Comparison with robust ensemble-based methods.**
> * If we understand correctly, we assume that you might be referring to [1], which is able to orthogonally use in our approach. We have carried out additional experiments with ensemble-based NAS. We estimate the robustness by ensembling zero-cost proxies on the NAS-Bench-201 search space across CIFAR-10, CIFAR-100, and ImageNet16-120. As depicted in Table, our proxy exhibits an improved correlation. This suggests that the robust evaluation for given neural architectures can be significantly enhanced by ensemble integration with our proxy. (Detailed table can be found in Table R6 in PDF)
> | |CIFAR-10||||CIFAR-100||||ImageNet16-120|||
> |-|-|-|-|-|-|-|-|-|-|-|-|
> ||Clean|FGSM|PGD|CC|Clean|FGSM|PGD|CC|Clean|FGSM|PGD|
> |CRoZe|0.823|0.826|0.188|0.719|0.784|0.786|0.343|0.765|0.765|0.596|**0.707**|
> |Ensemble|0.803|0.681|0.543|0.771|0.793|0.739|**0.444**|0.798|0.749|0.638|0.296|
> |CRoZe+Ensemble|**0.894**|**0.872**|**0.633**|**0.894**|**0.894**|**0.851**|0.415|**0.878**|**0.810**|**0.688**|0.259|
>
> [1]Adelfattah et al., Zero-Cost Proxies for Lightweight NAS, ICLR 2021

---

> > ### Author Response · Authors · 2023-08-17
> > **Gentle Reminder**
> >
> > Dear Reviewer,
> >
> > This is a gentle reminder for the discussion period.
> >
> > We have addressed all of your initial concerns and provide additional experimental results in our previous responses.
> >
> > We sincerely hope to discuss our work and look forward to your constructive comments. For your convenience, we provide the **short summary** of our previous responses to your initial concerns. Detailed responses can be found in the previous responses.
> >
> > Best,\
> > Author
> >
> > ---
> >
> > > ### Summary
> > >- **Evaluation of adversarially trained models against recent adversarial attacks on NAS-Bench-201 search space and DARTS search space on CIFAR-10.**
> > >   - (Shown through additional experiment in Table R1) Our method achieves **the highest overall correlation of 0.399 for average robust accuracies (CW, DeepFool, SPSA, LGV, and AutoAttack)** while the best baseline shows 0.352.
> > >   - (Shown through additional experiment in Table R2, Figure R1) Our proxy demonstrates **the highest average robustness of 55.77%** against various attacks (CW, LGV, SPSA, and AutoAttack) with **15 times less search cost** compared to baseline.
> > >- **Explanation on how CRoZe can handle novel types of perturbations.**
> > >   - Our proxy identifies architectures that maintain consistent features, gradients and parameters across diverse perturbations, making them suitable for any type of perturbations. (Please visit our initial response with a detailed explanation.)
> > >- **The reason for using DARTS search space.**
> > >   - Because 1) existing baselines use DARTS search space, and 2) it is the largest search space for image classification tasks.
> > >- **Compatibility with ensemble-based NAS.**
> > >   - (Shown through additional experiment in Table R6) We demonstrate that our proxy can improve the correlation for clean and robust accuracies on diverse tasks when our proxy is incorporated to ensemble-based NAS on NAS-Bench-201 search space.

---

> > > ### Author Response · Authors · 2023-08-20
> > > **Gentle Reminder**
> > >
> > > Dear Reviewer,
> > >
> > > **With just 1 day left in our discussion period**, we politely ask you to review our recent responses and the attached file showcasing the additional experimental results you requested. For your convenience, **a summary also has been provided in our previous response.**
> > >
> > > We truly appreciate your commitment to the review process and await your valuable feedback.
> > >
> > > Warm regards,\
> > > Author

---

> > > > ### Author Response · Authors · 2023-08-21
> > > > **Final Reminder**
> > > >
> > > > Dear reviewer,
> > > >
> > > > **With only a few hours left in the discussion period, we kindly remind you to review our response.**\
> > > > We respectfully request that you check our response within this timeframe.\
> > > > If there are any remaining concerns, please let us know.
> > > >
> > > > Best,
> > > > Author

---

> > ### Comment · Reviewer_m6P2 · 2023-08-21
> >
> > I have carefully reviewed the initial submission, the authors' response, and the feedback from other reviewers. I appreciate the effort that has been invested in addressing the concerns raised, and I would like to thank the authors for exhaustive set of experimental results. The responses provide answerers to my questions, and in light of this, I have updated my rating accordingly.

---

> > > ### Author Response · Authors · 2023-08-21
> > > **Thanks for your reply**
> > >
> > > Dear Reviewer,
> > >
> > > We sincerely appreciate the time and effort you've dedicated to reviewing our work.\
> > > Your positive comments and detailed concerns have greatly enhanced the quality of our work.
> > >
> > > Best regards,\
> > > Author

---

### Author Rebuttal · Authors · 2023-08-10

Dear Reviewers,

We deeply appreciate the time and effort you have invested in reviewing our paper. During the initial response period, we did our best to address all the concerns you raised in the response. Moreover, **we have thoughtfully included the additional experimental results that you requested in the attached PDF file**. We kindly request your thorough consideration of our responses and the additional experiments in the PDF. For your convenience, we have provided tables summarizing the contents, corresponding to each reviewer. We hope that our revisions have fully resolved your concerns, and we kindly request that you consider reflecting these changes in your updated review scores.

Best,\
Authors

---
### **Summary**
> **Reviewer m6P2**
> - Evaluation of adversarially trained models against recent adversarial attacks (Table R1, Table R2, Figure R1)
> - Compatibility with ensemble-based NAS (Table R6)
---
> **Reviewer DQhw**
> - Evaluation with different weight initialization methods (Table R3)
> - Influence of different types of adversarial training (Response)
> - End-to-end NAS performance against diverse adversarial attacks (Table R2)
---
> **Reviewer 41nW**
> - Evaluation against diverse adversarial attacks (Table R1)
> - Comparison with human-made architectures (Table R4)
> - Evaluation against AutoAttack (Table R2)
> - Verification the generalizability to diverse perturbations (Table R1, R2, 1, 2, 3)
---
> **Reviewer KEHv**
> - Comparison with recent train-free NAS works (Table R5)
> - Experimental results on final clean and robust accuracy (Table R2, 3)
---
### **Contents in PDF file**
* **[Table R1]**: Evaluation of adversarially trained models against diverse adversarial attacks on NAS-Bench-201 search space
   * Reviewer m6P2 (Weakness 1,2)
   * Reviewer 41nW (Weakness 1,2)
* **[Table R2/Figure R1]**: End-to-end performance of adversarially trained models against diverse adversarial attacks on DARTS search space
   * Reviewer m6P2 (Weakness 2)
   * Reviewer DQhw (Weakness 2)
   * Reviewer 41nW (Weakness 3)
* **[Table R3]**: Comparisons with diverse weight initialization methods of our proxy
   * Reviewer DQhw (Weakness 1)
* **[Table R4]**: Comparisons with human-made architectures
   * Reviewer 41nW (Weakness 3)
* **[Table R5]**: Comparisons with more recent train-free NAS baselines
   * Reviewer m6P2 (Question 3)
* **[Table R6]**: Compatibility with ensemble-based NAS
   * Reviewer KEHv (Weakness 2)

---

### Author Response · Authors · 2023-08-15
**Gentle Reminder: Responses are Uploaded**

**Dear Reviewers,**

Thanks for your time and effort for reviewing our paper. We politely ask you to go over our responses and the attached PDF file since we have only 1 week left for the discussion period that we can have interactions with you.

We have answered all your comments, and provided additional experimental results that you have requested. Therefore, please review our responses and reflect these in your final review and score.

We sincerely hope to discuss our work with you and look forward to your insightful and constructive comments.

Thanks,\
Authors

---

### Comment · Area_Chair_N51Q · 2023-08-19
**Response request for reviewers to authors' response.**

Dear Reviewer m6P2, 41nW and KEHv for NeurIPS submission 2479,

This is a friendly reminder for reading authors' response and responding to it accordingly.

Thank you,
Your AC

---

### Comment · Area_Chair_N51Q · 2023-08-21
**Final reminder to answer reviewers to authors' response.**

Dear Reviewer m6P2, 41nW and KEHv for NeurIPS submission 2479,

This is a final reminder for reading authors' response and responding to it accordingly.

Thank you,
Your AC

---

### Decision · Program_Chairs · 2023-09-21

**Decision:**

Accept (poster)

**Comment:**

The paper receives mostly positive reviews acknowledging novelty of the approach of zero cost NAS robuestness and the authors promptly respond to reviewers' questions. However, the reviewer KEHv's comments for better presentation and less thorough empirical validation are still remaining. Accordingly, the AC recommends to accept the submission to NeurIPS 2023 but strongly recommends to apply reviewer KEHv's comments in the final camera ready version.